# PARPs and ADP-Ribosylation in Chronic Inflammation: A Focus on Macrophages

**DOI:** 10.3390/pathogens12070964

**Published:** 2023-07-23

**Authors:** Diego V. Santinelli-Pestana, Elena Aikawa, Sasha A. Singh, Masanori Aikawa

**Affiliations:** 1Center for Interdisciplinary Cardiovascular Sciences, Division of Cardiovascular Medicine, Department of Medicine, Brigham and Women’s Hospital, Harvard Medical School, Boston, MA 02115, USA; dsantinellipestana@partners.org (D.V.S.-P.); eaikawa@bwh.harvard.edu (E.A.); sasingh@bwh.harvard.edu (S.A.S.); 2Center for Excellence in Vascular Biology, Division of Cardiovascular Medicine, Department of Medicine, Brigham and Women’s Hospital, Harvard Medical School, Boston, MA 02115, USA; 3Channing Division of Network Medicine, Department of Medicine, Brigham and Women’s Hospital, Harvard Medical School, Boston, MA 02115, USA

**Keywords:** immunity, mass spectrometry, proteomics, ADP-ribosylation, poly(ADP-ribose) glycohydrolase, Diphtheria toxin-like ADP-ribosyltransferases, chronic infection, arboviruses, cardiovascular disease, emphysema, alcoholic liver disease, SARS-CoV-2, host–pathogen interactions

## Abstract

Aberrant adenosine diphosphate-ribose (ADP)-ribosylation of proteins and nucleic acids is associated with multiple disease processes such as infections and chronic inflammatory diseases. The poly(ADP-ribose) polymerase (PARP)/ADP-ribosyltransferase (ART) family members promote mono- or poly-ADP-ribosylation. Although evidence has linked PARPs/ARTs and macrophages in the context of chronic inflammation, the underlying mechanisms remain incompletely understood. This review provides an overview of literature focusing on the roles of PARP1/ARTD1, PARP7/ARTD14, PARP9/ARTD9, and PARP14/ARTD8 in macrophages. PARPs/ARTs regulate changes in macrophages during chronic inflammatory processes not only via catalytic modifications but also via non-catalytic mechanisms. Untangling complex mechanisms, by which PARPs/ARTs modulate macrophage phenotype, and providing molecular bases for the development of new therapeutics require the development and implementation of innovative technologies.

## 1. Introduction

Poly(ADP-ribose) polymerases (PARPs), or ADP-ribosyltransferases (ARTs), catalyze the covalent transfer of ADP-ribose (ADPr) groups from NAD+ onto target biological macromolecules including nucleic acids (DNA, mRNA), transcription factors (e.g., NF-κB), or enzymes (e.g., PARP1/ARTD1 auto-ADP-ribosylation) [1]. The process of adding a single ADPr moiety is known as mono-ADP-ribosylation (MARylation), whereas adding multiple ADPr moieties is known as poly-ADP-ribosylation (PARylation); the latter occurs in a sequential way, starting with the transfer of one ADPr unit followed by the transfer of additional ADPr units onto a growing chain. The PARP enzyme family comprises 17 members, with variable functionality. PARP1/ARTD1, PARP2/ARTD2, PARP5a/TNKS1/ARTD5, and PARP5b/TNKS2/ARTD6 are poly-ARTs and have PARylation activity, while PARP3/ARTD3, PARP4/ARTD4, PARP6/ARTD17, PARP7/ARTD14, PARP8/ARTD16, PARP9/ARTD9, PARP10/ARTD10, PARP11/ARTD11, PARP12/ARTD12, PARP14/ARTD8, PARP15/ARTD7, and PARP16/ARTD15 are mono-ARTs and have MARylation activity [2]. PARP13/ARTD13 is catalytically inactive [3]. The poly-ARTs’ catalytic activity is counterbalanced by poly(adenosine diphosphate-ribose)-glycohydrolase (PARG) that hydrolyzes PARylation to MARylation. PARG is completely unable to hydrolyze the MAR covalently attached to proteins [4], however, the biological significance of this limitation remains unclear [5]. Nonetheless, this PARG enzymatic property is convenient for mass spectrometry-enabled ribosylome profiling (more below), since only the MARylated form of the modification is conducive to mass spectrometric analysis [6]. The mono-ARTs’ catalytic activity can be counterbalanced by enzymes other than PARG, such as ADP-ribosylhydrolase 3 (ARH3) [7,8], terminal ADP-ribose protein glycohydrolase (TARG)/C6orf130 [9,10], and MacroD1 [11] and MacroD2 [12], which are able to hydrolyze the MAR attached to proteins, functioning as mono-ADP-ribosylhydrolases. PARPs/ARTs also orchestrate biological processes via non-catalytic activities, such as directly binding to nuclear DNA or binding to transcription factors, but these roles remain to be further explored [13].

The subcellular locations of the PARPs/ARTs also dictate their biological functions. In a comprehensive analysis of human somatic cell lineages, Vyas et al. used N-terminal green fluorescent protein (GFP) and affinity-purified peptide antibodies to study the cellular localization of PARPs/ARTs and the occurrence of PARylation during the cell cycle [14]. PARP1/ARTD1 localized to the nucleus; PARP5a/TNKS1/ARTD5, PARP5b/TNKS2/ARTD6, PARP12/ARTD12, PARP13/ARTD13, PARP6/ARTD17, PARP8/ARTD16, PARP10/ARTD10, and PARP16/ARTD15 localized to the cytoplasm; and PARP2/ARTD2, PARP3/ARTD3, PARP7/ARTD14, PARP9/ARTD9, PARP14/ARTD8, PARP4/ARTD4, and PARP11/ARTD11 localized to the nucleus and cytoplasm. Their findings also suggested that: firstly, the expression of most PARPs/ARTs was pervasive across human tissues; secondly, while PARPs/ARTs could be found in the nucleus and in the cytoplasm, they were predominantly found in the cytoplasm; thirdly, PAR levels were influenced by the cell cycle, and the proportion of PAR identified in the nucleus versus cytoplasm changed during the cell cycle. Leung et al. demonstrated that PARP5a/TNKS1/ARTD5, PARP12/ARTD12, two isoforms of PARP13/ARTD13, PARP14/ARTD8, and PARP15/ARTD7 coordinate the assembly of stress granules in the cytoplasm, modifying each other within this cellular compartment [15]. Ryu et al. demonstrated that low concentrations of NAD+ can limit PARP1/ARTD1 activity in the nucleus [16]. The influx or efflux of NAD+ thus interferes with PARP1/ARTD1′s activity, leading to alternative gene expression signatures in the early process of adipogenesis [17]. Additionally, the predominant type of ADP-ribosylation in distinct cellular compartments seems to vary: PARylation appears to occur primarily in the nucleus [18], whereas MARylation in the cytoplasm [19]. These examples illustrate that PARPs/ARTs’ functions vary based on ADP-ribosylation catalytic activity, cellular compartment location, and the physiological and/or pathological microenvironment.

PARP/ART biology has been studied in the context of the innate immune system, with a particular focus on macrophages [20]. PARPs/ARTs were associated with biological responses mediated by IFN-γ, TNF-α, IL-1β, IL-6, and NF-κB, such as host–pathogen interactions in viral infections, vascular inflammation, and others [20]. In 1985, Singh et al. generated DNA double-strand breaks to induce PARylation in human monocytes [21]; and in 1991, Berton et al. reported that PARylation levels increased after IFN-γ stimulation in human macrophages [22]. More recently, Heer et al. demonstrated that the catalytic activities of PARP7/ARTD14, PARP10/ARTD10, PARP12/ARTD12, and PARP14/ARTD8 were closely connected to nicotinamide and derivates in the establishment of cellular innate immune response during COVID-19 infection [23]. In another pathological setting, Wang et al. reported that PARP1/ARTD1 and PARP2/ARTD2 inhibition with olaparib [24,25,26,27] (a PARP1/ARTD1/2 inhibitor approved by the Food and Drug Administration for the treatment of ovarian, breast, pancreatic, and prostate cancer) induced macrophage reprogramming towards an anti-tumor, pro-inflammatory phenotype [28]. Macrophages are found within the microenvironments of solid tumors [29] and chronic inflammatory conditions such as diabetes [30], neurodegenerative diseases [31], prolonged bacterial infections [32] but exhibit distinct functions. In tumors, they are associated with an anti-tumor innate immune response [33] but are hypothesized to promote and sustain a pro-inflammatory tissue milieu [20]. To date, research has focused primarily on the role of PARPs/ARTs in cancer biology, whereas their roles in other macrophage-driven pathologies are only just beginning to be explored.

As we will highlight further below, recent studies have begun to investigate more the role(s) of PARPs/ARTs and ADP-ribosylation in macrophage activation, with the aim to identify therapeutic avenues for acute and chronic inflammation. Olaparib is already available for the treatment of cancer and is also being investigated as a potential therapy for pulmonary arterial hypertension (Clinical Trial No.: NCT03782818), although these applications focus on DNA damage and repair features of PARP1/ARTD1 inhibition. Nevertheless, a phase-I trial has been initiated to test the PARP14/ARTD8 inhibitor, RBN-3143, as a potential therapy for atopic dermatitis (Clinical Trial No.: NCT05215808), focusing on inflammation control. Macrophages are key players in the sustained inflammation occurring in atopic dermatitis [34,35], thus representing the initiative to translate the interplay between ADP-ribosylation and macrophage biology to the clinic.

## 2. PARylation, Macrophages, and Chronic Inflammation

Among the PARPs/ARTs mediating PARylation during inflammation, PARP1/ARTD1 is the most studied [36,37]. Various stimuli promote PARP1/ARTD1 activity in macrophages, often leading to the expression of pro-inflammatory genes and downstream inflammatory responses. In the current section, we will review the most recent articles exploring the distinct mechanisms of action of PARP1/ARTD1 in macrophage activity in the setting of chronic or prolonged inflammation.

### 2.1. ADP-Ribosylation and DNA Damage

The catalytic activity of PARP1/ARTD1 increases with DNA damage following genotoxic stimuli. Dawicki-McKenna et al. used hydrogen/deuterium exchange–mass spectrometry to demonstrate that breaks in the DNA strand led to structural changes in PARP1/ARTD1′s helical subdomain (HD), which is part of the catalytic domain [38]. The helical subdomain functions as an autoinhibitory portion of the catalytic domain, unfolding in the presence of DNA strand breaks and thus promoting PARP1/ARTD1′s catalytic activity. Eustermann et al. [39] demonstrated that a sequential multidomain unfolding occurs in PARP1/ARTD1 in response to DNA single-strand breaks (SSBs). Firstly, the F2 domain recognizes and detects SSBs; secondly, the F1 domain binds to the complex, exposing the 5′ cryptic site and orienting the assembly of remaining PARP1/ARTD1 domains; thirdly, the F3, WGR, and CAT domains also bind the exposed strand, culminating in the unfolding of the autoinhibitory helical subdomain. This cooperative process generates a specific recognition of sites of SSBs by PARP1/ARTD1, promoting PAR-mediated signaling and modulation of chromatin structure upon DNA damage. Figure 1 provides a graphical representation [40] of PARP1/ARTD1 domains and their structure, as well as a flowchart indicating the dual action of PARP1/ARTD1 during inflammation.

These works also aided in the paradox involving PARP1/ARTD1 *cis* versus *trans* (another PARP1/ARTD1 molecule) modification during response to DNA damage. While PARP1/ARTD1 dimers have been reported [41,42,43], suggesting that the *trans* modification occurs, results from Dawicki-McKenna et al. [38] and Eustermann et al. [39] indicated that PARP1/ARTD1 automodifies itself, unless two DNA binding sites are closely adjacent, leading to *trans* modification activity.

More recently, other reports have described the dynamic nature of the interactions between PARP1/ARTD1 and DNA [44], either using its DNA-binding domain (DBD) along with zinc finger domains I and II (ZI and ZII, respectively) for short-term interactions [45], or using histone H4, which leads to a prolonged interaction with the DNA strand [46]. Short-term interactions between the DBD of PARP1/ARTD1 and DNA were associated with activation of DNA repair pathways at specific stages of DNA damage, while long-term interactions between the C-terminal domain of PARP1/ARTD1 and histone H4 were associated with promotion of gene expression [44]. This dual action of PARP1/ARTD1 on DNA illustrates the complexity of this enzyme and provides indications that PARP1/ARTD1 may be associated with chronic inflammation not only as a repair mechanism secondary to inflammation-driven DNA damage [47] but also promoting the expression of pro-inflammatory and/or anti-inflammatory genes.

Reactive oxygen species (ROS) generated during pro-inflammatory responses lead to DNA damage [48], triggering short-term PARP1/ARTD1-DNA interactions [44,45]. However, in a study using a model of elastase-induced emphysema and chronic lung inflammation in mice, prolonged inhibition of PARP1/ARTD1 with olaparib reduced the number of macrophages in the bronco-alveolar lavage after 21 days of treatment when compared with the control group [49]. Levels of ROS and malondialdehyde (MDA, a marker of lipid peroxidation) increased in lung tissues of the control group (four-fold and seven-fold, respectively) due to the inflammation and macrophage activity induced by elastase, but daily treatment with olaparib restored ROS and MDA to normal levels, indicating an improvement in the inflammatory and redox balances [49,50]. These results exemplify how PARP1/ARTD1 may have a dual and contrasting role in chronic inflammation, repairing DNA following ROS while promoting the production of ROS in macrophages.

### 2.2. PARP1/ARTD1 Promotes Transcription of Pro-Inflammatory and Apoptosis-Related Genes

Inhibition of PARP1/ARTD1 ameliorates inflammation in chronic conditions and innate immune responses, and this effect was found in multiple pathologies driven by long-term inflammatory processes. Kunze et al. [51] demonstrated that stimulation of bone marrow-derived monocytes (BMDMs) from genetically modified mice expressing catalytically inactive PARP1/ARTD1 induced the expression of a pro-inflammatory signature of almost 2500 genes, including genes regulating IL-12, IFN-γ, and TNF-α production. In the same study, they reported that mice transplanted with catalytically inactive PARP1/ARTD1 myeloid progenitors were colonized by H. pylori at higher levels when compared to their control littermates [51], suggesting that PARP1/ARTD1 contributed to controlling gastric bacterial colonization. In another disease model, inhibition of PARP1/ARTD1 with 3-aminobenzamide, an anti-inflammatory compound classically used for PARP1/ARTD1 inhibition [52], improved rectal hemorrhage, blood sugar levels, blood IL-1β levels, weight loss, and the histological score of colonic sections in mice with colitis-associated diabetes [53]. Similar findings were reported by Kovács et al. [54] after using olaparib to inhibit PARP1/ARTD1 activity in a mouse model of Crohn’s disease (a type of inflammatory bowel disease). They found that olaparib increased the levels of IL-10, while it suppressed the concentration of IL-1β and IL-6 [54]. Also, olaparib generated a reduction in the number of monocytes in the blood of treated mice when compared with controls [54]. Gupte et al. [55] stimulated BMDMs from wild-type and PARP1/ARTD1-deficient mice, demonstrating that PARP1/ARTD1-mediated STAT1-α PARylation influenced the transcriptional program upon IFN-γ stimulation [55].

The regulation of PARPs/ARTs’ catalytic activities in chronic inflammation also relates to NAD+ metabolism. Gerner et al. [56] inhibited nicotinamide phosphoribosyltransferase (NAMPT), a rate-limiting enzyme in the NAD+ salvage pathway, to reduce NAD+ levels in human cells and mice with intestinal colitis. They found that depletion of NAD+ reduced PARP1/ARTD1 catalytic activity, suppressed the expression of IL-6, IL-1β, and TNF-α, and skewed monocytes/macrophages from pro-inflammatory towards anti-inflammatory phenotypes [56]. In the same line, reduction of NAMPT-derived NAD+ via pharmacological inhibition of NAMPT reduced the pathological changes in psoriasis [57] and atopic dermatitis [58] and diminished the expression of pro-inflammatory biomarkers.

In addition to promoting cytokine/chemokine gene expression, PARP1/ARTD1 also influences the cellular fate in apoptosis [59], a fundamental element of inflammation [60]. PARP1/ARTD1 has been extensively associated with caspases in a mechanism known as parthanatos [61,62] (not reviewed in this manuscript). For instance, Zhang et al. [63] analyzed cleaved caspase 3 in liver samples from mice with chronic alcoholic liver injury [63], and demonstrated that pharmacological inhibition with PJ-34 [64] or genetic depletion of PARP1/ARTD1 decreased the number of cleaved caspase 3-positive cells in diseased livers when compared to controls. They found that long-term ethanol consumption promoted PARP/ART activation, hepatic steatosis, and intense cytokine expression in liver samples, while in vivo pharmacologic inhibition of PARP1/ARTD1 with PJ-34 attenuated triglyceride content and serum alanine transaminase levels in liver, suggesting a milder injury phenotype [63]. Erener et al. [65] also identified an association between caspase 1, caspase 7, and PARP1/ARTD1. They found that stimulation with LPS promoted the translocation of caspase 7 to the nucleus (mediated by caspase 1 and NLRP3 inflammasome activation), where it cleaves PARP1/ARTD1 at the caspase cleavage site D214, generating free PARP1/ARTD1 fragments, decondensation of chromatin, and expression of NF-κB dependent-genes. They generated human THP-1 cells expressing non-cleavable PARP1/ARTD1, stimulated them with LPS, and compared them with genetically unmodified controls, confirming that caspase 7 cleaved PARP1/ARTD1 mostly at the D214 site. Martínez-Morcillo et al. [66] found that PARP1/ARTD1 activation leads to skin inflammation and cell death via parthanatos-mediated apoptosis in psoriasis, and pharmacological inhibition of NAMPT decreased the expression of genes associated with psoriasis.

Together, those findings suggest that PARP1/ARTD1 can influence gene expression during chronic inflammation via ADP-ribosylation of macromolecules and can initiate apoptosis upon interaction with caspases. Controlling NAD+ levels via NAMPT regulation in such immune responses may be a potential source of new targets to suppress pathogenesis derived from ADP-ribosylation, although a deeper understanding of these mechanisms is still needed.

### 2.3. PARP1/ARTD1 Mediates Host–Pathogen Interactions in Chagas Heart Disease

Chagas heart disease is caused by the protozoan parasite *Trypanosoma cruzi* (*T. cruzi*). The classical phenotype seen in this condition is the result of chronic (years to decades) of sustained myocyte inflammation, oxidative stress, and macrophage infiltration into cardiac muscle [67,68]. Ba et al. [69] demonstrated that *T. cruzi* infection of cardiomyocytes leads to mitochondrial production of ROS that diffuse to the cytosol and nucleus, leading to DNA damage and PARP1/ARTD1 activation. As a result, the expression of genes related to pro-inflammatory cytokines increased either due to the interaction between ROS and cytosolic NF-κB or due to PARP1/ARTD1-mediated PARylation of proteins that interact with RelA(p65) (an NF-κB subunit). Further evidence indicated that depletion of PARP1/ARTD1 (with genetic deletion or PJ-34 administration) in infected mice prevented cardiac hypertrophy and left ventricle dysfunction and restored the mitochondrial antioxidant/oxidant balance [70]. PARP1/ARTD1 is associated with chromatin during *T. cruzi* infection but its mRNA levels did not change when compared to non-infected states, indicating that a translocation of PARP1/ARTD1 to chromatin-dense regions occurred [71,72]. These results suggest that PARP1/ARTD1 influences the response to mitochondrial stress during *T. cruzi* infection. Evidence also connects PARP1/ARTD1 to macrophages in the host–pathogen interaction. Macrophage-like RAW264.7 cells treated with extracellular vesicles (EVs) derived from infected mouse plasma released higher levels of TNF-α, IL-1β, and IL-6 than did control cells [73]. EVs derived from *T. cruzi*-infected RAW264.7 cells induced lower expression levels of TNF-α, IL-1β, and IL-6 in BMDMs harvested from PARP1/ARTD1-deficient mice compared to wild-type control [73]. Thus, it is possible that the previously described role of macrophages in Chagas heart disease [74] may be mediated by PARP1/ARTD1, but more studies are needed.

### 2.4. PARP1/ARTD1 in Cardiovascular Inflammation

Von Lukowicz et al. proposed that PARP1/ARTD1 mediates macrophage adhesion to endothelial cells in the process of atherogenesis [75]. Both PARP1/ARTD1 and PARP2/ARTD2 inhibition with PJ-34 and PARP1/ARTD1 genetic deletion without PJ-34 reduced plaque formation and the expression of adhesion molecules such as E-selectin, P-selectin, VCAM1, and iNOS. Another study linked high glucose and PARP1/ARTD1 levels in streptozocin-induced diabetes mellitus in apolipoprotein E-deficient mice [76]. In a rat model of cerebral aneurysms, treatment with 3-aminobenzamide, an anti-inflammatory compound classically used for PARP1/ARTD1 inhibition [52], decreased macrophage accumulation and PARP1/ARTD1 expression [77]. These studies indicate that different forms of inflammatory arterial injury (i.e., atherosclerosis, aneurysm formation, and hyperglycemia-induced inflammation) share PARP1/ARTD1 as a common mediator of the inflammatory process.

## 3. MARylation, Macrophages, and Chronic Inflammation

Although most studies to date have focused on PARylation and PARP1/ARTD1, evidence suggests that MARylation also regulates macrophage activation, inflammation, and host–pathogen interactions. For instance, in an evolutionary analysis of the PARP/ART genes, Daugherty et al. [78] demonstrated that PARP9/ARTD9, PARP14/ARTD8, and PARP15/ARTD7 had signs of genetic adaptation in primates, notably in their macrodomains, and evolved under positive selection. In another example, our own research demonstrated that PARP9/ARTD9 and PARP14/ARTD8 regulate pro-inflammatory activation of macrophages upon stimulation [79,80]. Therefore, considering the accumulated evidence that MARylation and mono-PARPs/ARTs are involved with the innate immune system, in this section we will review recent articles investigating the interplay between PARPs/ARTs, MARylation, and macrophage activation and explore how these findings provide insight into mechanisms that drive chronic inflammation and host–pathogen interactions. 

### 3.1. PARP7/ARTD14 Mediates Epithelial Inflammation

In a mouse model of a dextran sodium sulfate-induced ulcerative colitis study, PARP7/ARTD14 deletion increased mRNA levels of IL-1β, IL-6, IL-17, and Lcn2 and decreased survival rate [81]. Aryl hydrocarbon receptor (AHR), which induces the expression of PARP7/ARTD14, mediates pro-inflammatory responses in this model. The PARP7/ARTD14 catalytic domain MARylates AHR, which represses AHR signaling in a negative feedback loop. AHR responsiveness was enhanced by short-chain fatty acids in mouse colonocytes [81], supporting the hypothesis that chronic inflammation related to toxic lipid particles in cells of epithelial origin involves PARP7/ARTD14 [82].

### 3.2. PARP9/ARTD9 Mediates Viral and Bacterial Host–Pathogen Interactions

In a cohort with patients infected with pulmonary tuberculosis (TB) and healthy controls, Chen et al. [83] identified an inversely proportional association between TB infection severity and methylation status of PARP9/ARTD9 DNA in PBMCs extracted from participants. Severe TB clinical phenotypes were associated with hypomethylation of the PARP9/ARTD9 gene, suggesting that lower expression of PARP9/ARTD9 may lead to impaired innate response to TB infection in individuals with that epigenotype. Novel data from Thirunavukkarasu et al. [84] further support this hypothesis. They reported that PARP9/ARTD9 mRNA was increased in humans and mice infected with TB, and *Parp9-/-* mice were more susceptible to TB infection and developed more severe phenotypes compared to controls.

Similarly, PARP9/ARTD9 appears to be involved in innate immune responses against RNA viruses. Xing et al. [85] demonstrated that PARP9/ARTD9 is able recognize and bind RNA virus in human and mouse dendritic cells and macrophages, deploying an IFN-mediated response independent of the mitochondrial anti-viral signaling (a major mechanism for recognizing RNA viruses during infection). Furthermore, *Parp9-/-* deletion made mice more susceptible to RNA virus infection [85], reinforcing that PARP9/ARTD9 participates in the host–pathogen interactions. Curiously, PARP9/ARTD9 was associated with persistent hepatitis B virus (HBV) infection in a transcriptome-wide association study, in which chronic HBV carriers had increased expression of PARP9/ARTD9 when compared to non-infected individuals [86]. HBV is a DNA virus with unique features that approximate it to RNA viruses [87], which may relate to the results above (PARP9/ARTD9 acting as a recognizer of viral RNA).

### 3.3. PARP14/ARTD8 Mediates Chronic Inflammation and Response to Arboviruses

Recent data indicate that PARP14/ARTD8 participates in the establishment of an immune response to arboviruses. Eckei et al. [88] reported that the macrodomains of Chikungunya virus (a positive single-strand RNA virus) have strong hydrolase activity on proteins that were ADP-ribosylated by PARP10/ARTD10, PARP14/ARTD8, and PARP15/ARTD7. Fernandez et al. [89] reported that Zika virus infection in human PBMCs induced the expression of PARP14, IL-6, CCL8, CXCL1, and CXCL5, suggesting that the infection promoted changes in the transcriptional and post-transcriptional levels. These results indicate that PARP14/ARTD8 influences the host–pathogen dynamic in arbovirus infections.

### 3.4. PARP9/ARTD9 and PARP14/ARTD8 Mediate Macrophage Activation in Atherosclerosis

PARP14/ARTD8 is also important in other chronic inflammatory responses. Using a systems approach based on unbiased network analysis and artificial intelligence, our previous studies discovered PARP14/ARTD8 and PARP9/ARTD9 as potential molecular switches of macrophage activation [79,80]. Proteome analyses from stimulated and non-stimulated human and mouse macrophage-like cells detected an increase in the ADP-ribosylated PARP14/ARTD8 and PARP9/ARTD9 peptide levels upon stimulation with IFN-γ, and network analysis identified a close link between those PARPs/ARTs and the human coronary artery disease gene module [79,80]. Additional in vitro experiments indicated that PARP9/ARTD9 and PARP14/ARTD8 may function upstream of pro-inflammatory STAT1 and anti-inflammatory STAT6 signaling pathways, respectively [79,80]. Iqbal et al. [90] reported that macrophages from PARP14/ARTD8-deficient mice express higher levels of tissue factor mRNA and protein than do wild-type mice [90]. Mehrotra et al. [91] reported that PARP14/ARTD8 specifically binds to STAT6, regulating its promoter activity upon stimulation with IL-4, and demonstrated that this interaction is dependent on PARP14/ARTD8 catalytic domain [91]. Figure 2 provides a summary of the different disease models mediated by PARPs/ARTs and macrophages in chronic inflammation.

## 4. SARS-CoV-2, ADP-Ribosylation, and Innate Immune Response

ADP-ribosylation and PARPs/ARTs are important in viral host–pathogen interactions and in the organization of the host’s innate immune response [92]. PARP/ART genes are interferon-stimulated genes [93,94]. Stress granule formation is a major effect of interferon stimulation during innate immune responses, and it is closely related to PARPs/ARTs and ADP-ribosylation [95,96,97]. PARPs/ARTs and PARG probably mediate the assembly and maintenance of stress granules in a dynamic way: PARylation of stress granule proteins increases in stress conditions or with PARG silencing [15,98]. Together, these results suggest that hydrolysis of PAR/MAR could limit the effectiveness of the host’s innate immune response against viruses.

Even before the severe acute respiratory syndrome coronavirus (SARS-CoV)-2 pandemic, different types of coronaviruses’ macrodomains belonging to non-structural protein 3 (nsp3) were identified as ADP-ribose-binding modules [99]. In 2006, Egloff et al. reported that the crystal structure of the SARS-CoV macrodomain associates with ADP-ribose, being able to bind PAR and to function as an ADP-ribose 1”-phosphatase [100]. Evidence suggests that nsp3 and its macrodomains were part of coronaviruses’ virulence mechanisms [101,102,103], promoting virus replication and suppressing interferon-mediated host responses (e.g., stress granule formation) [104,105,106].

With the onset of the pandemic, data connecting the new SARS-CoV-2 macrodomains and ADP-ribosylation quickly became available [107]. The crystal structures of SARS-CoV-2 nsp3 and its macrodomains were the initial focus of many research groups, often associating structural studies [108,109,110,111] and computational methods [112,113] to identify potential treatments for the infection. Alhammad et al. [114] reported that SARS-CoV-2 nsp3 macrodomain 1 (Mac1) hydrolyzes MARylated proteins, functioning as a mono-ADP-ribosylhydrolase. This macrodomain’s function is preserved across the three coronaviruses that caused pandemics in the recent past: SARS-CoV, SARS-CoV-2, and Middle East respiratory syndrome (MERS) coronavirus. Brosey et al. [115] compared human PARG with Mac1 crystal structures and identified homology between their active sites, revealing that PARG inhibitor fragments PARG-345 and PARG-329 can fully interact with Mac1, appearing as potential inhibitors for Mac1. Chea et al. [116] proposed that Mac1 has specific targets and functionality when compared to Mac2 and Mac3. Their results indicate that Mac1 may act specifically in the ADP-ribose moieties on O- and N-linked groups, being able to cleave ADP-ribosylated substrates via a-NAD+, ADPr-1”phosphate, and O-acetyl-ADP-ribose, but not via b-NAD+, a-ADP-ribose-(arginine), and ADP-ribose-(serine)-histone H3.

Other studies also investigated the link between Mac1 and innate immunity responses against SARS-CoV-2. Russo et al. [117] demonstrated that ectopic nsp3 (macrodomain not specified) is able to hydrolyze downstream ADP-ribosylation mediated by PARP9-DTX3L dimers following IFN-γ stimulation. Preliminary data indicated [118] that deletion of Mac1 in SARS-CoV-2 (ΔMac1) led to a faster clearance of the virus in a mouse model of severe infection when compared to wild-type SARS-CoV-2. ΔMac1 also promoted the expression of ISGs and interferons and sharply reduced the number of inflammatory neutrophils and macrophages.

There is another mechanism by which SARS-CoV-2 may intervene in the host–pathogen interaction via ADP-ribosylation. PARP/ART catalytic activity depends on NAD+ for the covalent transference of ADP-ribose to biological macromolecules. It is also well established that increased PARP/ART catalytic activity leads to depletion of NAD+ [119,120]. Reports before SARS-CoV-2 already suggested that restoration of NAD+ would enhance host immune responses against viruses, aiding macrophage function and the interferon cascade [121]. Based on this background, authors hypothesized that NAD+ may also be a key element of pathogenesis in acute and chronic (post-acute sequelae of COVID-19) SARS-CoV-2 infection [120,122,123]. Heer et al. [23] demonstrated that varied human lung cell lineages infected with SARS-CoV-2 have increased expression of PARP7/ARTD14, PARP10/ARTD10, PARP12/ARTD12, and PARP14/ARTD8 (among other PARPs/ARTs) and that NAD+ concentration was the limiting factor for these enzymes. In addition, the authors of the same study demonstrated that infection of human cells with murine hepatitis virus (a model of coronaviruses) leads to NAD+ and NADP+ depletion and that SARS-CoV-2 changes the expression of genes related to NAD biosynthesis [23]. Using SARS-CoV-2-infected mice, Jiang et al. [124] confirmed that SARS-CoV-2 infection alters the expression of genes related to NAD and NADPH biosynthesis. They also demonstrated that NAD+ supplementation alleviated the pathological phenotypes of pneumonia in infected mice and partially rescued the imbalance in NAD+ genes. Table 1 provides an overview of the enzymes discussed in this review, with their respective activities and disease processes.

## 5. Mass Spectrometry and ADP-Ribosylation

Enzyme-catalyzed covalent modifications of amino acids such as phosphorylation, ubiquitination, glycosylation, and ADP-ribosylation are post-translational modifications (PTMs). PTMs regulate various processes related to cellular homeostasis [125]. The biological complexity and the potentially ephemeral nature of PTMs make them challenging to decipher; but innovative mass spectrometry technologies have enabled their widespread investigations. Ribosylomics is the study of proteome-wide ADP-ribosylation, using mass spectrometry. The last ten years have witnessed technological advances that have overcome initial obstacles for ribosylome, including difficulties associated with precise identification of amino acid acceptor sites, the unstable nature of its covalent binding to the amino acid chain, and its complex pattern of fragmentation [126,127,128,129,130,131,132]. Several mass spectrometry-based workflows are currently available to study ADP-ribosylation, but PARylated peptides are not amenable to typical mass spectrometric acquisition methods. The conversion of PAR to MAR peptides using poly-PARG [80,131,132] or the complete reduction to a phosphoribose using a phosphodiesterase [133] provides the means to detect ADP-ribosylated proteins/peptides using mass spectrometry; however, these methods do not provide the nature of the original modification, MAR vs. PAR. In parallel to the mass spectrometry-based innovations to characterize ADP-ribosylated proteins are the ongoing efforts to develop computational resources to confidently characterize and report ribosylome data.

### 5.1. Enrichment Strategies and Activation Methods Influence the Identification of ADP-Ribosylated Proteins in Macrophages

Multiple research groups have tested the activation methods for identifying ADP-ribosylated peptides, their ADP-ribose acceptor sites, and unique enrichment strategies. Electron transfer dissociation (ETD) proved to be efficient in the identification of unambiguous ADP-ribosylated peptides and their acceptor sites [127], with the combination of ETD with higher-energy collisional dissociation (EThcD) being superior to ETD alone for the same purpose [128,132]. Also, the enrichment protocol using an Af1521-Sepharose bead workflow [134] can be combined with ETD for mass spectrometry analysis [131]. Different activation methods may provide the identification of specific ADP-ribosylated peptide groups, depending on their acceptor sites. Ion ETD seems to be superior to EthcD in the occurrence of non-dissociative electron transfer for ADP-ribosylated precursor peptides, and residues modified on arginine and lysine were more stable during HCD fragmentation, whereas the annotation of residues modified on serine, glutamate, tyrosine, and aspartate were more challenging [126]; this is interesting, as modifications on arginine were more frequent during physiological conditions, while modifications on serine were scarce in similar conditions [131], indicating that adjusting the activation method based on the biological condition may provide more reliable results. It is worth noting that the studies mentioned in this paragraph mainly used cancer cell lines and/or healthy mouse tissues, indicating that technical optimization would potentially be needed for the study of inflammatory biosystems related to macrophages and macrophage-like cells.

In 2019, our group demonstrated that different ADP-ribosylation enrichment strategies and activation methods influence the identification of ADP-ribosylated peptides in IFN-γ-stimulated human THP-1 macrophage-like cells [80]. We compared the mass spectrometry results of two enrichment strategies: the Af1521-based workflow [134], in which the macrodomain of the Af1521 compound is used to affinity-purify ADP-ribosylated peptides; and the 10H anti-ADP-ribose antibody-based workflow, in which the antibody is used in immunoprecipitation of ADP-ribosylated proteins. While the Af1521-based workflow provided spectra rich in MARylated peptides and amino acid acceptor sites, the antibody-based workflow only provides peptides suggesting candidate ADP-ribosylated proteins since ADP-ribosylated peptides themselves are not detected. The 10H strategy enriched 1,389 candidate ADP-ribosylated proteins, whereas the Af1521 strategy enriched 145 ADP-ribosylated proteins, resulting in 39 proteins commonly identified which included PARP14/ARTD8 and PARP9/ARTD9 [80].

We also compared distinct activation (peptide sequencing) methods and demonstrated that, while HCD provides a larger number of identified ADP-ribosylated peptides, ETD dissociation provides a more reliable identification of the ADP-ribosylation acceptor site in ADP-ribosylated peptides [80]. With these results, we were able to confirm that stimulation of human THP-1 macrophage-like cells with IFN-γ increased PARP9/ARTD9 and PARP14/ARTD8 ADP-ribosylation levels.

### 5.2. An Innovative Spectral Annotation Strategy Facilitates the Report of ADP-Ribosylated Peptides in IFN-γ-Stimulated Mice

The investigation of ADP-ribosylated proteins using mass spectrometry methods requires optimal annotation strategies to accurately identify such proteins after enrichment protocols. The ADPriboDB (initially published by Vivelo et al. [135] and updated by Ayyappan et al. [136]) was the first report of a publicly available database of ADP-ribosylated proteins, in which users can find information about proteins reported in the mass spectrometry literature, from as early as 1975. Each individual entry was revised manually by two independent reviewers before inclusion in the database. Likewise, manual annotation of peptide spectra is still commonly used in mass spectrometry studies in the ADP-ribosylation field.

In 2022, our group capitalized on our optimized ADP-ribosylation enrichment and activation methods to develop a new strategy for annotating ADP-ribosylated peptide spectra (named “RiboMap”) from liver and spleen samples of IFN-γ-stimulated mice [129,137]. In this strategy, once a candidate ADP-ribosylated peptide fragment spectrum is assigned and scored by the standard spectral search engine, RiboMaP then annotates and scores the spectra for ADP-ribosylation-unique features [129].

With this unique spectral annotation tool, we could increase the confidence of the reported ADP-ribosylated peptide spectra associated with pro-inflammatory responses in liver and spleen. With that combination of mass spectrometry and computational techniques, even ADP-ribosylated peptides with overall low biological abundances, such as PARP9/ARTD9 and PARP14/ARTD8, could be identified. We further applied the RiboMap strategy to publicly available data sets and even to our own previously published human macrophage cell and mouse samples, and we found that, regardless of study and sample type, RiboMap increased the number of ADP-ribosylated peptide spectral annotations in all tests [129].

## 6. Future Perspectives

Since the seminal articles in the 1960s describing ADP-ribosylation as a post-translational modification occurring in stimulated human monocytes/macrophages [21,22,138], the field has expanded enormously. Although ADP-ribosylation is posited to regulate various biological or pathological processes, the mechanisms remain barely understood. Mass spectrometry and computational biology techniques appear to be some of the fundamental tools for overcoming knowledge gaps in the study of ADP-ribosylation. Mass spectrometry technologies are continually developing, with the aim to sequence deeper into a proteome. Ion mobility technology, such as field asymmetric waveform ion mobility spectrometry (FAIMS), is one such development. We envision FAIMS to be a promising technology to increase the sequencing depth of a given ribosylome, similar to what has been demonstrated for the phosphorylation field [139].

There are also promising perspectives for the field of ADP-ribosylation and macrophage-mediated chronic inflammation. A clinical trial is currently investigating the efficacy of a PARP1/ARTD14 inhibitor compound in the treatment of atopic dermatitis, a chronic inflammatory disease deeply associated with macrophage activation. Likewise, with the help of the strategies mentioned above, other novel drug targets may emerge from bench research. The cumulative evidence suggests that ADP-ribosylation is an important piece of the sustained inflammatory response of macrophages in cardiovascular, gastrointestinal, pulmonary, and hepatic diseases, as well as in prolonged infections. Therefore, we expect that other potential candidate drugs may appear in a near future, translating bench results into clinical tools for patient care.

## Figures and Tables

**Figure 1 pathogens-12-00964-f001:**
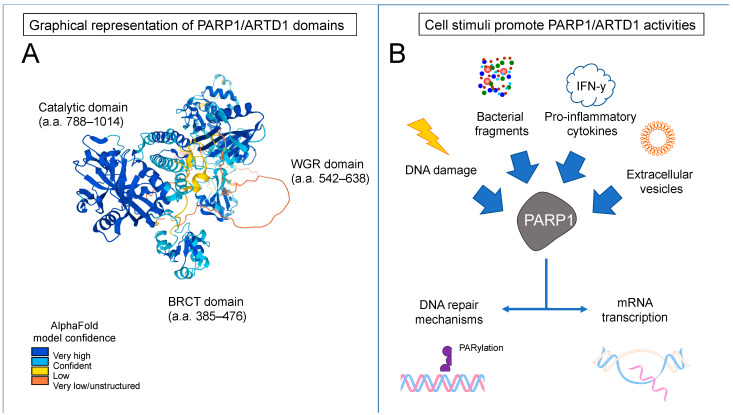
Graphical representation of PARP1/ARTD1 and its role during inflammation. (**A**): Tridimensional representation of PARP1/ARTD1 structure, indicating the catalytic, the WGR, and the BRCT domains. (**B**): Flowchart depicting the cell stimuli able to promote PARP1/ARTD1 activity (catalytic and non-catalytic) during prolonged inflammatory processes, leading to PARylation and transcription of mRNA.

**Figure 2 pathogens-12-00964-f002:**
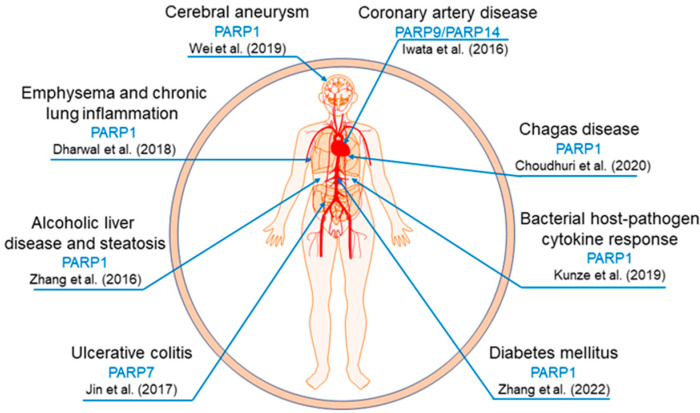
Representation of the disease models influenced by the interplay between PARPs/ARTs and macrophages in chronic inflammation [49,51,63,73,76,77,79,81].

**Table 1 pathogens-12-00964-t001:** PARPs/ARTs and ADP-ribose hydrolases and their action in chronic inflammation.

Enzyme	Activity	Disease/Biological Process(s)
PARP1/ARTD1	PARylation, MARylation, or non-catalytic activity	Emphysema/Chronic lung inflammation [49,50] H. pylori infection [51] Colitis/Inflammatory bowel diseases [53,54,56] Psoriasis [57,66] Atopic dermatitis [58] Alcoholic liver injury [63] Chagas heart disease [69,70,71,72]
PARP7/ARTD14	MARylation	SARS-CoV-2 infection [23] Colitis/Inflammatory bowel diseases [53,54,56]
PARP9/ARTD9	MARylation	Pulmonary tuberculosis [83,84] RNA-viruses infections [85,86] Atherosclerosis/arterial inflammation [79,80] SARS-CoV-2 infection [117]
PARP10/ARTD10	MARylation	SARS-CoV-2 infection [23] Arboviruses infections [88]
PARP12/ARTD12	MARylation	Assembly and maintenance of stress granules [15] SARS-CoV-2 infection [23]
PARP13/ARTD13	Non-catalytic activity	Assembly and maintenance of stress granules [15]
PARP14/ARTD8	MARylation	Atopic dermatitis (Clinical Trial No.: NCT05215808) SARS-CoV-2 infection [23] Atherosclerosis/arterial inflammation [79,80,90,91] Arboviruses infections [88,89]
PARP15/ARTD7	MARylation	Assembly and maintenance of stress granules [15] Arboviruses infections [88]
PARG	Hydrolysis (PAR)	Assembly and maintenance of stress granules [15,98]
Macrodomain 1 (nsp3)	Hydrolysis (MAR)	SARS-CoV-2 infection [114,117,118]

## Data Availability

No new data were created or analyzed in this study. Data sharing is not applicable to this article.

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
