# Peer review of "PARPs and ADP-Ribosylation in Chronic Inflammation: A Focus on Macrophages"

_pathogens, 2023, doi:10.3390/pathogens12070964_

Round 1
Reviewer 1 Report
The authors should mention more erasers of adp-ribose
such as TARG and ARH3. Also to be up to date with the literature the nomenclature of Luscher on PARPs should be cited. The Ribo Map method to identify sites of ADP-ribosylation) should include the names of
target proteins. Finally the recent on SARS COV-2 bearing a macro domain capable of binding ADP-ribose) should be discussed.
Author Response
Response to Reviewer #1
Comment: “The authors should mention more erasers of adp-ribose such as TARG and ARH3. Also to be up to date with the literature the nomenclature of Luscher on PARPs should be cited. The Ribo Map method to identify sites of ADP-ribosylation should include the names of target proteins. Finally the recent on SARS COV-2 bearing a macro domain capable of binding ADP-ribose should be discussed.”
Response: Thank you for your valuable comments and suggestions. As suggested, we updated the nomenclature used, according to the consensus published by Luscher et al.
Regarding the Ribo Map excerpt, we used PARP9/ARTD9 and PARP14/ARTD8 as examples of ADP-ribosylated proteins identified with the help of Ribo Map, as shown below.
Page 17; lines 516-524:
“With this unique spectral annotation tool, we could increase the confidence of the reported ADP-ribosylated peptide spectra associated with pro-inflammatory responses in liver and spleen. With that combination of mass spectrometry and computational techniques, even ADP-ribosylated peptides with overall low biological abundances, such as PARP9/ARTD9 and PARP14/ARTD8, could be identified. We further applied the RiboMap strategy to publicly available data sets and even to our own previously published human macrophage cell and mouse samples, and we found that, regardless of study and sample type, RiboMap increased the number of ADP-ribosylated peptide spectral annotation in all tests [129].”
Regarding the action of ADP-ribose erasers, we included them into our Introduction while explaining the overview of ADP-ribose writers and erasers.
Page 2, Lines 43-56:
“The poly-ARTs catalytic activity is counterbalanced by poly-(adenosine diphosphate-ribose)-glycohydrolase (PARG) that hydrolyzes PARylation to MARylation. PARG is completely unable to hydrolyze the MAR covalently attached to proteins [4], however the biological significance of this limitation remains unclear [5]. Nonetheless, this PARG enzymatic property is convenient for mass spectrometry-enabled ribosylome profiling (more below), since only the MARylated form of the modification is conducive to mass spectrometric analysis [6]. The mono-ARTs’ catalytic activity can be counterbalanced by enzymes other than PARG, such as ADP-ribosylhydrolase 3 (ARH3) [7,8], Terminal ADP-ribose protein Glycohydrolase (TARG)/C6orf130 [9,10], and MacroD1 [11] and MacroD2 [12], which are able to hydrolyze the MAR attached to proteins, functioning as mono-ADP-ribosylhydrolases. PARPs/ARTs also orchestrate biological processes via non-catalytic activities, such as directly binding to nuclear DNA or binding to transcription factors, but these roles remain to be further explored [13].”
In addition, we created a new section in our manuscript, exploring the relationship between ADP-ribosylation and SARS-CoV-2 infection, in which we vastly discuss the role of SARS-CoV-2 macrodomains as ADP-ribose erasers, as shown below:
Pages 12-13, lines 364 -427:
“4. SARS-CoV-2, ADP-ribosylation and innate immune response
ADP-ribosylation and PARPs/ARTs are important in viral host-pathogen interactions and in the organization of host’s innate immune response [92]. PARPs/ARTs genes are interferon-stimulated genes [93,94]. Stress granule formation is a major effect of interferon stimulation during innate immune responses, and it is closely related to PARPs/ARTs and ADP-ribosylation [95-97]. PARPs/ARTs and PARG probably mediate the assembly and maintenance of stress granules in a dynamic way: PARylation of stress granule proteins increases in stress conditions or with PARG silencing [15,98]. Together, these results suggest that hydrolysis of PAR/MAR could limit the effectiveness of the host’s innate immune response against viruses.
Even before the severe acute respiratory syndrome coronavirus (SARS-CoV)-2 pandemic, different types of coronaviruses’ macro domains belonging to the nonstructural protein 3 (nsp3) were identified as ADP-ribose binding modules [99]. In 2006, Egloff et al. reported that the crystal structure of SARS-CoV macro domain associates with ADP-ribose, being able to bind PAR and to function as an ADP-ribose 1”-phosphatase [100]. Evidence suggests that nsp3 and its macro domains were part of coronaviruses’ virulence mechanisms [101-103], promoting virus replication and suppressing interferon-mediated host responses (e.g., stress granules formation) [104-106].
With the onset of the pandemic, data connecting the new SARS-CoV-2 macro domains and ADP-ribosylation quickly became available [107]. The crystal structures of SARS-CoV-2 nsp3 and its macro domains were the initial focus of many research groups, often associating structural studies [108-111] and computational methods [112,113] to identify potential treatments for the infection. Alhammad et al. [114] reported that SARS-CoV-2 nsp3 macro domain 1 (Mac1) hydrolyzes MARylated proteins, functioning as a mono-ADP-ribosylhydrolase. This macro domain function is preserved across the three coronavirus that caused pandemics in a recent past: SARS-CoV, SARS-CoV-2, and Middle East respiratory syndrome coronavirus (MERS). Brosey et al. [115] compared human PARG with Mac1 crystal structures and identified homology between their active sites, revealing that PARG inhibitor fragments PARG-345 and PARG-329 can fully interact with Mac1, posing as potential inhibitors for Mac1. Chea et al. [116] proposed that Mac1 has specific targets and functionality when compared to Mac2 and Mac3. Their results indicate that Mac1 may act specifically in the ADP-ribose moieties on O- and N-linked groups, being able to cleave ADP-ribosylated substrates via a-NAD+, ADPr-1”phosphate, and O-acetyl-ADP-ribose, but not via b-NAD+, a-ADP-ribose-(arginine), and ADP-ribose-(serine)-histone H3.
Other studies also investigated the link between Mac1 and innate immunity responses against SARS-CoV-2. Russo et al. [117] demonstrated that ectopic nsp3 (macro domain not specified) is able to hydrolyze downstream ADP-ribosylation mediated by PARP9-DTX3L dimers following IFN-γ stimulation. Preliminary data indicated [118] that deletion of Mac1 in SARS-CoV-2 (ΔMac1) led to a faster clearance of the virus in a mouse model of severe infection when compared to wild-type SARS-CoV-2. ΔMac1 also promoted the expression of ISGs and interferons, and sharply reduced the number of inflammatory neutrophils and macrophages.
There is another mechanism by which SARS-CoV-2 may intervene in the host-pathogen interaction via ADP-ribosylation. PARP/ART catalytic activity depends on NAD+ for the covalent transference of ADP-ribose to biological macromolecules. It is also well established that increased PARP/ART catalytic activity leads to depletion of NAD+ [119,120]. Reports before SARS-CoV-2 already suggested that restoration of NAD+ would enhance host immune responses against viruses aiding macrophage function and the interferon cascade [121]. Based on this background, authors hypothesized that NAD+ may also be a key element of pathogenesis in acute and chronic (post-acute sequelae of COVID-19) SARS-CoV-2 infection [120,122,123]. Heer et al. [23] demonstrated that varied human lung cell lineages infected with SARS-CoV-2 have increased expression of PARP7/ARTD14, PARP10/ARTD10, PARP12/ARTD12, and PARP14/ARTD8 (among other PARPs/ARTs), and that NAD+ concentration was the limiting factor for these enzymes. In addition, the authors of the same study demonstrated that infection of human cells with murine hepatitis virus (a model of coronaviruses) lead to NAD+ and NADP+ depletion, and that SARS-CoV-2 changes the expression of genes related to NAD biosynthesis [23]. Using SARS-CoV-2 infected mice, Jiang et al. [124] confirmed that SARS-CoV-2 infection alters the expression of genes related to NAD and NADPH biosynthesis. They also demonstrated that NAD+ supplementation alleviated the pathological phenotypes of pneumonia in infected mice, and partially rescued the unbalance in NAD+ genes.”
Reviewer 2 Report
The mini review by Masanori Aikawa et all was focused on macrophage activation mediated by PARPs during chronic inflammatory processes; they made a honest work, but it is opinion of this referee that it does not perfectly fit the scope of the special issue "ADP-Ribosylation in Pathogens" being not focused on pathogens. It does not comply the guest editors requests: “…we are calling for the community to submit original research articles or reviews on elucidating the role of ADP-ribosylation in pathogen biology, pathogenesis, and host-pathogen interactions, as well as their potential avenues for drug development”.
The same authors already published an exhaustive review “The impact of PARPs and ADP-ribosylation on inflammation and host–pathogen interactions” (Genes and Development 34:341-359, 2020) focused on the required topic, that could have been used as background for an update based on recent papers such as: The Critical Role of PARPs in Regulating Innate Immune Responses by Huifang Zhu et al Frontiers 2021; Virus–Host Interplay Between Poly (ADP-Ribose) Polymerase 1 and Oncogenic Gamma herpes viruses Woo-Chang Chung et al Frontiers 2022; The Antiviral Activities of Poly-ADP-Ribose Polymerases Mathilde Malgras et al 2021)
Author Response
Response Reviewer #2
Comment: “The mini review by Masanori Aikawa et all was focused on macrophage activation mediated by PARPs during chronic inflammatory processes; they made a honest work, but it is opinion of this referee that it does not perfectly fit the scope of the special issue "ADP-Ribosylation in Pathogens" being not focused on pathogens. It does not comply the guest editors requests: “…we are calling for the community to submit original research articles or reviews on elucidating the role of ADP-ribosylation in pathogen biology, pathogenesis, and host-pathogen interactions, as well as their potential avenues for drug development”.
The same authors already published an exhaustive review “The impact of PARPs and ADP-ribosylation on inflammation and host–pathogen interactions” (Genes and Development 34:341-359, 2020) focused on the required topic, that could have been used as background for an update based on recent papers such as: The Critical Role of PARPs in Regulating Innate Immune Responses by Huifang Zhu et al Frontiers 2021; Virus–Host Interplay Between Poly (ADP-Ribose) Polymerase 1 and Oncogenic Gamma herpes viruses Woo-Chang Chung et al Frontiers 2022; The Antiviral Activities of Poly-ADP-Ribose Polymerases Mathilde Malgras et al 2021).”
Response: Thank you for your valuable comments and suggestions. We carefully evaluated our manuscript based on your comments, and we made the following modifications.
#1. As mentioned in your comment, we went back to our previous review, and used it as a background for updates. It was written/published before the global installation of the SARS-CoV-2 pandemic. We thus created a whole new section recapitulating the investigation of SARS-CoV-1, MERS, and SARS-CoV-2, and their connection with ADP-ribosylation in the host-pathogen interaction. We believe that readers will now benefit from understanding the background of ADP-ribosylation and coronaviruses (discussed in our previous review article), and the development of new discoveries based on SARS-CoV-2. Please, see below.
Pages 12-13, lines 364 -427:
“4. SARS-CoV-2, ADP-ribosylation and innate immune response
ADP-ribosylation and PARPs/ARTs are important in viral host-pathogen interactions and in the organization of host’s innate immune response [92]. PARPs/ARTs genes are interferon-stimulated genes [93,94]. Stress granule formation is a major effect of interferon stimulation during innate immune responses, and it is closely related to PARPs/ARTs and ADP-ribosylation [95-97]. PARPs/ARTs and PARG probably mediate the assembly and maintenance of stress granules in a dynamic way: PARylation of stress granule proteins increases in stress conditions or with PARG silencing [15,98]. Together, these results suggest that hydrolysis of PAR/MAR could limit the effectiveness of the host’s innate immune response against viruses.
Even before the severe acute respiratory syndrome coronavirus (SARS-CoV)-2 pandemic, different types of coronaviruses’ macro domains belonging to the nonstructural protein 3 (nsp3) were identified as ADP-ribose binding modules [99]. In 2006, Egloff et al. reported that the crystal structure of SARS-CoV macro domain associates with ADP-ribose, being able to bind PAR and to function as an ADP-ribose 1”-phosphatase [100]. Evidence suggests that nsp3 and its macro domains were part of coronaviruses’ virulence mechanisms [101-103], promoting virus replication and suppressing interferon-mediated host responses (e.g., stress granules formation) [104-106].
With the onset of the pandemic, data connecting the new SARS-CoV-2 macro domains and ADP-ribosylation quickly became available [107]. The crystal structures of SARS-CoV-2 nsp3 and its macro domains were the initial focus of many research groups, often associating structural studies [108-111] and computational methods [112,113] to identify potential treatments for the infection. Alhammad et al. [114] reported that SARS-CoV-2 nsp3 macro domain 1 (Mac1) hydrolyzes MARylated proteins, functioning as a mono-ADP-ribosylhydrolase. This macro domain function is preserved across the three coronavirus that caused pandemics in a recent past: SARS-CoV, SARS-CoV-2, and Middle East respiratory syndrome coronavirus (MERS). Brosey et al. [115] compared human PARG with Mac1 crystal structures and identified homology between their active sites, revealing that PARG inhibitor fragments PARG-345 and PARG-329 can fully interact with Mac1, posing as potential inhibitors for Mac1. Chea et al. [116] proposed that Mac1 has specific targets and functionality when compared to Mac2 and Mac3. Their results indicate that Mac1 may act specifically in the ADP-ribose moieties on O- and N-linked groups, being able to cleave ADP-ribosylated substrates via a-NAD+, ADPr-1”phosphate, and O-acetyl-ADP-ribose, but not via b-NAD+, a-ADP-ribose-(arginine), and ADP-ribose-(serine)-histone H3.
Other studies also investigated the link between Mac1 and innate immunity responses against SARS-CoV-2. Russo et al. [117] demonstrated that ectopic nsp3 (macro domain not specified) is able to hydrolyze downstream ADP-ribosylation mediated by PARP9-DTX3L dimers following IFN-γ stimulation. Preliminary data indicated [118] that deletion of Mac1 in SARS-CoV-2 (ΔMac1) led to a faster clearance of the virus in a mouse model of severe infection when compared to wild-type SARS-CoV-2. ΔMac1 also promoted the expression of ISGs and interferons, and sharply reduced the number of inflammatory neutrophils and macrophages.
There is another mechanism by which SARS-CoV-2 may intervene in the host-pathogen interaction via ADP-ribosylation. PARP/ART catalytic activity depends on NAD+ for the covalent transference of ADP-ribose to biological macromolecules. It is also well established that increased PARP/ART catalytic activity leads to depletion of NAD+ [119,120]. Reports before SARS-CoV-2 already suggested that restoration of NAD+ would enhance host immune responses against viruses aiding macrophage function and the interferon cascade [121]. Based on this background, authors hypothesized that NAD+ may also be a key element of pathogenesis in acute and chronic (post-acute sequelae of COVID-19) SARS-CoV-2 infection [120,122,123]. Heer et al. [23] demonstrated that varied human lung cell lineages infected with SARS-CoV-2 have increased expression of PARP7/ARTD14, PARP10/ARTD10, PARP12/ARTD12, and PARP14/ARTD8 (among other PARPs/ARTs), and that NAD+ concentration was the limiting factor for these enzymes. In addition, the authors of the same study demonstrated that infection of human cells with murine hepatitis virus (a model of coronaviruses) lead to NAD+ and NADP+ depletion, and that SARS-CoV-2 changes the expression of genes related to NAD biosynthesis [23]. Using SARS-CoV-2 infected mice, Jiang et al. [124] confirmed that SARS-CoV-2 infection alters the expression of genes related to NAD and NADPH biosynthesis. They also demonstrated that NAD+ supplementation alleviated the pathological phenotypes of pneumonia in infected mice, and partially rescued the unbalance in NAD+ genes.”
#2. We also shifted the focus of our manuscript towards host-pathogen interactions. For that, we created new subsections and expanded previously existing subsections to resonate the focus on host-pathogen interactions. Please, see below the new/modified subsections dedicated to discussing host-pathogen interactions (in addition to the new SARS-CoV-2 section showed above). We believe that these modifications will help readers to have a better understanding of ADP-ribosylation and host-pathogen interactions in prolonged/chronic inflammation.
Page 8, lines 241-264 :
“2.3. PARP1/ARTD1 mediates host-pathogen interactions in Chagas Heart Disease
Chagas Heart Disease is caused by the protozoan parasite Trypanosoma cruzi (T. cruzi). The classical phenotype seen in this condition is the result of chronic (years to decades) of sustained myocyte inflammation, oxidative stress, and macrophage infiltration into cardiac muscle [67,68]. Ba et al. [69] demonstrated that T. cruzi infection of cardiomyocytes leads to mitochondrial production of ROS that diffuse to the cytosol and nucleus leading to DNA damage and PARP1/ARTD1 activation. As a result, the expression of genes related to pro-inflammatory cytokines increased either due to the interaction between ROS and cytosolic NF-kB or due to PARP1/ARTD1-mediated PARylation of proteins that interact with RelA(p65) (an NF-kB subunit). Further evidence indicated that depletion of PARP1/ARTD1 (with genetic deletion or PJ-34 administration) in infected mice prevented cardiac hypertrophy and left ventricle dysfunction and restored the mitochondrial antioxidant/oxidant balance [70]. PARP1/ARTD1 associated with chromatin during T. cruzi infection but its mRNA levels did not change when compared to non-infected states, indicating that a translocation of PARP1/ARTD1 to chromatin dense regions occurred [71,72]. These results suggest that PARP1/ARTD1 influences the response to mitochondrial stress during T. cruzi infection. Evidence also connects PARP1/ARTD1 to macrophages in the host-pathogen interaction. Macrophage-like RAW264.7 cells treated with extracellular vesicles (EVs) derived from infected mice plasma released higher levels of TNF-α, IL-1β, and IL-6 than did control cells [73]. EVs derived from T. cruzi-infected RAW264.7 cells induced lower expression levels of TNF-α, IL-1β, and IL-6 in BMDMs harvested from PARP1/ARTD1-deficient mice compared to wildtype control [73]. Thus, it is possible that the previously described role of macrophages in Chagas Heart Disease [74] may be mediated by PARP1/ARTD1, but more studies are needed.”
Pages 10-11, lines 299-344:
“3.2. PARP9/ARTD9 mediates viral and bacterial host-pathogen interactions
In a cohort with patients infected with pulmonary tuberculosis (TB) and healthy controls, Chen et al. [83] identified an inversely proportional association between TB infection severity and methylation status of PARP9/ARTD9 DNA in PBMCs extracted from participants. Severe TB clinical phenotypes associated with hypomethylation of PARP9/ARTD9 gene, suggesting that lower expression of PARP9/ARTD9 may lead to impaired innate response to TB infection in individuals with that epigenotype. Novel data from Thirunavukkarasu et al. [84] further support this hypothesis. They reported that PARP9/ARTD9 mRNA was increased in humans and mice infected with TB, and Parp9-/- mice were more susceptible to TB infection and developed more severe phenotypes when compared to controls.
Similarly, PARP9/ARTD9 appears to be involved in innate immune responses against RNA viruses. Xing et al. [85] demonstrated that PARP9/ARTD9 is able recognize and bind RNA virus in human and mice dendritic cells and macrophages, deploying an IFN-mediated response independent of the mitochondrial antiviral-signaling (a major mechanism for recognizing RNA viruses during infection). Furthermore, Parp9-/- deletion made mice more susceptible to RNA virus infection [85], reinforcing that PARP9/ARTD9 participates in the host-pathogen interactions. Curiously, PARP9/ARTD9 was associated with persistent Hepatitis B virus (HBV) infection in a transcriptome-wide association study, in which chronic HBV carriers had increased expression of PARP9/ARTD9 when compared to non-infected individuals [86]. HBV is a DNA virus with unique features that approximate it to RNA viruses [87], which may relate to the results above in this paragraph (PARP9/ARTD9 acting as a recognizer of viral RNA).
3.3 PARP14/ARTD8 mediates chronic inflammation and response to arboviruses
Recent data indicate that PARP14/ARTD8 participates in the establishment of an immune response to arboviruses. Eckei et al. [88] reported that the macrodomains of Chikungunya virus (a positive single strand RNA virus) have strong hydrolase activity on proteins that were ADP-ribosylated by PARP10/ARTD10, PARP14/ARTD8 and PARP15/ARTD7. Fernandez et al. [89] reported that Zika virus infection in human PBMCs induced the expression of PARP14, IL-6, CCL8, CXCL1, and CXCL5, suggesting that the infection promoted changes in the transcriptional and post-transcriptional levels. These results indicate that PARP14/ARTD8 influences the host-pathogen dynamic in arboviruses infections.”
Reviewer 3 Report
In this article, Santinelli-Pestana review the role of ADP-ribosylation in chronic inflammatory processes, with a focus on PARPs in macrophages. I find the review to be rather poorly written, with many factual imprecisions and an insufficient balance, excessively highlighting a select number of studies without integrating them into the larger body of literature. The three core issues I would like to see addressed are described below:
1. Factual imprecisions and incorrect definitions and nomenclature are frequent, particularly in the introduction, and need to be thoroughly revised. Some examples (which are not comprehensive):
- “poly ADP polymerase enzymes (PARPs)” (line13-14) – should read “poly(ADP-ribose) polymerase”
- “PARPs catalyse the binding of ADP-ribose groups“ (line 27-28) – the ADP-ribose groups are transferred covalently onto targets
- “organic molecules” (line 28) – perhaps “biological macromolecules” would be more appropriate
- “adding a chain of ADPr moieties” (line 31) – gives the false impression that a chain is assembled first and then added to a protein, when in fact one ADPr unit is added to the protein first, followed by transfer of additional ADPr units onto the growing chain
- “A PARP´s catalytic activity is counterbalanced by PARG” (lines 35-36); “PARPs and PARG work together in a delicate balance” (lines 41-42) – this is not the case for most PARPs, as they are mono-ADP-ribosyl transferases and therefore their products are not hydrolysed by PARG
- “PARG is unable to fully hydrolyze MAR bound to proteins” (line 37) – again, the MAR is not bound, but covalently attached to the protein, and PARG is actually completely unable to remove MARylation (Slade et al, Nature 2011)
- “the biological significance of this limitation is unclear” (line 38) - It is increasingly evident that other hydrolases, such as ARH3, TARG, MacroD1 and MacroD2 play important roles in MAR hydrolysis, and should be mentioned in the manuscript
- “Cambronne et al. demonstrated that low concentrations of NAD+ can limit PARP1 activity in the nucleus (9).” (lines 48-49) – This reference is a review article, perhaps the authors are referring to (Ryu et al., Science 2019)?
- “PARylation of DNA” (line 98) – although there is some evidence of direct DNA
PARylation, PARP1 is canonically considered to modify proteins
- “3-aminobenzamide, an anti-inflammatory compound” (line 205) – although not particularly specific, 3-AB is a classic PARP1 inhibitor (Durkacz et al, Nature, 1980)
2. The manuscript is insufficiently balanced, with excessive focus on a small number of studies, often without mention of other studies that much more comprehensively analysed a given phenomenon or that present conflicting evidence to the statement. Some examples (again not comprehensive):
- Lines 45 – 55: When referring to subcellular localization of PARPs, the study by (Vyas et al Nat Comms, 2013) should be discussed as it is a detailed and comprehensive analysis of the localization of the whole PARP family
- Lines 97-108: In a section titled “Rybosylation and DNA damage”, the canonical pathways of PARP1 activation by DNA breaks must first be thoroughly introduced (e.g. Dawicki-McKenna et al and Eustermann et al, both in Mol Cell 2015) before referring to a putative short term vs long term interaction of PARP1 with DNA recently identified in Drosophila (ref 31).
- Lines 282-317 are excessively focused on the mass spectrometry efforts of the authors, completely ignoring a huge body of literature on mass spectrometry analyses of ADP-ribosylation sites by many other groups.
3. Large parts of the manuscript are devoted to describing the results of individual papers, without integrating the knowledge gained from these studies into a coherent narrative, which is arguably the main purpose of a review manuscript. Examples: Lines 123 – 148 (about ref. 39); lines 149-162 (about ref.44); 163-186 (about ref. 45) and section 3.2.
Author Response
Response to Reviewer #3
Comment #1: “In this article, Santinelli-Pestana review the role of ADP-ribosylation in chronic inflammatory processes, with a focus on PARPs in macrophages. I find the review to be rather poorly written, with many factual imprecisions and an insufficient balance, excessively highlighting a select number of studies without integrating them into the larger body of literature. The three core issues I would like to see addressed are described below”.
Response: Thank you for valuable comments and suggestions. We reviewed our manuscript and promoted several changes based on your comments. Below you will find each comment/suggestion and the respective answer with the modification made to address your concerns. Please, see below.
Comment: Factual imprecisions and incorrect definitions and nomenclature are frequent, particularly in the introduction, and need to be thoroughly revised. Some examples (which are not comprehensive):
- “poly ADP polymerase enzymes (PARPs)” (line13-14) – should read “poly(ADP-ribose) polymerase”
Response: We updated the nomenclature used in this manuscript to be in accordance with the consensus published by Luscher et al (2021). The term “poly ADP polymerase enzymes” was substituted by “poly(ADP-ribose) polymerase” (page 1, line 30).
Comment: “PARPs catalyse the binding of ADP-ribose groups“ (line 27-28) – the ADP-ribose groups are transferred covalently onto targets
Response: The term “binding” was substituted in this and throughout the manuscript by the term(s) “covalent transfer”, as in: “Poly(ADP-ribose) polymerases (PARPs), or ADP-ribosyltransferases (ARTs), catalyze the covalent transfer of ADP-ribose (ADPr) groups from NAD+ onto target biological macromolecules” (page 1, lines 30-32)
Comment: “organic molecules” (line 28) – perhaps “biological macromolecules” would be more appropriate
Response: The term “organic molecules” was substituted by the term “biological macromolecules” (page 1, lines 31-32)
Comment: “adding a chain of ADPr moieties” (line 31) – gives the false impression that a chain is assembled first and then added to a protein, when in fact one ADPr unit is added to the protein first, followed by transfer of additional ADPr units onto the growing chain
Response: We re-wrote this part, incorporating your constructive description of the PARylation process:
Pages 1-2, lines 33-37:
“The process of adding a single ADPr moiety is known as mono-ADP-ribosylation (MARylation), whereas adding multiple ADPr moieties is known as poly-ADP-ribosylation (PARylation); the latter occurs in a sequential way, starting with the transfer of one ADPr unit followed by the transfer of additional ADPr units onto a growing chain.”
Comment: “A PARP´s catalytic activity is counterbalanced by PARG” (lines 35-36); “PARPs and PARG work together in a delicate balance” (lines 41-42) – this is not the case for most PARPs, as they are mono-ADP-ribosyl transferases and therefore their products are not hydrolysed by PARG
Response: We re-wrote this part, incorporating a clearer explanation of the dynamics involved in PARPs/ARTs and erasers such as PARG, MacroD1, MacroD2, etc:
Page 2, Lines 43-54:
“The poly-ARTs catalytic activity is counterbalanced by poly-(adenosine diphosphate-ribose)-glycohydrolase (PARG) that hydrolyzes PARylation to MARylation. PARG is completely unable to hydrolyze the MAR covalently attached to proteins [4], however the biological significance of this limitation remains unclear [5]. Nonetheless, this PARG enzymatic property is convenient for mass spectrometry-enabled ribosylome profiling (more below), since only the MARylated form of the modification is conducive to mass spectrometric analysis [6]. The mono-ARTs’ catalytic activity can be counterbalanced by enzymes other than PARG, such as ADP-ribosylhydrolase 3 (ARH3) [7,8], Terminal ADP-ribose protein Glycohydrolase (TARG)/C6orf130 [9,10], and MacroD1 [11] and MacroD2 [12], which are able to hydrolyze the MAR attached to proteins, functioning as mono-ADP-ribosylhydrolases.”
Comment: “PARG is unable to fully hydrolyze MAR bound to proteins” (line 37) – again, the MAR is not bound, but covalently attached to the protein, and PARG is actually completely unable to remove MARylation (Slade et al, Nature 2011)
Response: This suggestion was incorporated to the manuscript and was addressed in the excerpt above.
Page 2, lines 45-46:
“PARG is completely unable to hydrolyze the MAR covalently attached to proteins [4]”
Comment: “the biological significance of this limitation is unclear” (line 38) - It is increasingly evident that other hydrolases, such as ARH3, TARG, MacroD1 and MacroD2 play important roles in MAR hydrolysis, and should be mentioned in the manuscript
Response: This suggestion was incorporated to the manuscript and was addressed in the excerpt below.
Page 2, lines 50-54:
“The mono-ARTs’ catalytic activity can be counterbalanced by enzymes other than PARG, such as ADP-ribosylhydrolase 3 (ARH3) [7,8], Terminal ADP-ribose protein Glycohydrolase (TARG)/C6orf130 [9,10], and MacroD1 [11] and MacroD2 [12], which are able to hydrolyze the MAR attached to proteins, functioning as mono-ADP-ribosylhydrolases.”
Comment: “Cambronne et al. demonstrated that low concentrations of NAD+ can limit PARP1 activity in the nucleus (9).” (lines 48-49) – This reference is a review article, perhaps the authors are referring to (Ryu et al., Science 2019)?
Response: We substituted the previous reference (Cambrone et al) by the corresponding manuscript within the review article cited.
Page 3, lines 73-75:
“Ryu et al. demonstrated that low concentrations of NAD+ can limit PARP1/ARTD1 activity in the nucleus [16].”
Comment: “PARylation of DNA” (line 98) – although there is some evidence of direct DNA PARylation, PARP1 is canonically considered to modify proteins
Response: We re-wrote this part and avoided using the term “PARylation of DNA” and direct DNA PARylation, focusing on the mechanisms related to the interaction between PARP1/ARTD1 and DNA, as follows (please, see also another excerpt on Comment #2 regarding DNA activation by DNA damage):
Page 4, lines 124-125:
“The catalytic activity of PARP1/ARTD1 increases with DNA damage following genotoxic stimuli”.
Comment: “3-aminobenzamide, an anti-inflammatory compound” (line 205) – although not particularly specific, 3-AB is a classic PARP1 inhibitor (Durkacz et al, Nature, 1980)
Response: We incorporated this suggestion to the excerpt:
Page 6, lines 189-193:
“In another disease model, inhibition of PARP1/ARTD1 with 3-aminobenzamide, an anti-inflammatory compound classically used for PARP1/ARTD1 inhibition [52], improved rectal hemorrhage, blood sugar levels, blood IL-1β levels, weight loss, and the histological score of colonic sections in mice with colitis associated diabetes [53].”
Comment #2: “The manuscript is insufficiently balanced, with excessive focus on a small number of studies, often without mention of other studies that much more comprehensively analysed a given phenomenon or that present conflicting evidence to the statement. Some examples (again not comprehensive)”
Response: Thank you for this constructive feedback. We implemented several changes in the manuscript to address this issue, including creating new sections, creating new subsections, and re-writing parts that were overly focused on a small number of studies. Please, se the answers below.
Comment: Lines 45 – 55: When referring to subcellular localization of PARPs, the study by (Vyas et al Nat Comms, 2013) should be discussed as it is a detailed and comprehensive analysis of the localization of the whole PARP family
Response: We incorporated this suggestion and re-wrote the mentioned excerpt, thoroughly discussing the study by Vyas et al (2013).
Pages 2-3, lines 57-70:
“The sub cellular locations of the PARPs/ARTs also dictate their biological functions. In a comprehensive analysis of human somatic cell lineages, Vyas et al. used N-terminal green fluorescent protein and affinity-purified peptide antibodies to study the cellular localization of PARPs/ARTs and the occurrence of PARylation during the cell cycle [14]. PARP1/ARTD1 localized to the nucleus; PARP5a/TNKS1/ARTD5, PARP5b/TNKS2/ARTD6, PARP12/ARTD12, PARP13/ARTD13, PARP6/ARTD17, PARP8/ARTD16, PARP10/ARTD10, and PARP16/ARTD15 localized to the cytoplasm; and PARP2/ARTD2, PARP3/ARTD3, PARP7/ARTD14, PARP9/ARTD9, PARP14/ARTD8, PARP4/ARTD4, and PARP11/ARTD11 localized to the nucleus and cytoplasm. Their findings also suggested that: firstly, the expression of most PARPs/ARTs was pervasive across human tissues; secondly, while PARPs/ARTs could be found in the nucleus and in the cytoplasm, they were predominantly found in the cytoplasm; thirdly, PAR levels were influenced by the cell cycle, and the proportion of PAR identified in the nucleus versus cytoplasm changed during the cell cycle.”
Comment: Lines 97-108: In a section titled “Rybosylation and DNA damage”, the canonical pathways of PARP1 activation by DNA breaks must first be thoroughly introduced (e.g. Dawicki-McKenna et al and Eustermann et al, both in Mol Cell 2015) before referring to a putative short term vs long term interaction of PARP1 with DNA recently identified in Drosophila (ref 31). Dawicki-McKenna et al, Mol Cell 2015 Eustermann et al, Mol Cell 2015
Response: We incorporated this comment and discussed the suggested manuscripts into the section.
Page 4-6, lines 124-177:
“The catalytic activity of PARP1/ARTD1 increases with DNA damage following genotoxic stimuli. Dawicki-McKenna et al. used hydrogen/deuterium exchange-mass spectrometry to demonstrate that breaks in the DNA strand led to structural changes in PARP1/ARTD1’s helical subdomain (HD), which is part of the catalytic domain [38]. The helical subdomain functions as an autoinhibitory portion of the catalytic domain, unfolding in the presence of DNA strand breaks and thus promoting PARP1/ARTD1’s catalytic activity. Eustermann et al. [39] demonstrated that a sequential multidomain unfolding occurs in PARP1/ARTD1 in response to DNA single-strand breaks (SSB). Firstly, the F2 domain recognize and detects SSB; secondly, the F1 domains binds to the complex, exposing the 5’ cryptic site and orienting the assembly of remaining PARP1/ARTD1 domains; thirdly, the F3, WGR, and CAT domains also bind the exposed strand, culminating in the unfolding of the autoinhibitory helical subdomain. This cooperative process generates a specific recognition of sites of SSB by PARP1/ARTD1, promoting PAR-mediated signaling and modulation of chromatin structure upon DNA damage. Figure 1 provides a graphical representation [40] of PARP1/ARTD1 domains and their structure, as well as a flowchart indicating the dual action of PARP1/ARTD1 during inflammation.
These works also aided in the paradox involving PARP1/ARTD1 cis versus trans (another PARP1/ARTD1 molecule) modification during response to DNA damage. While PARP1/ARTD1 dimers have been reported [41-43], suggesting that the trans modification occurs, results from Dawicki-McKenna et al. [38] and Eustermann et al. [39] indicated that PARP1/ARTD1 auto modifies itself, unless two DNA binding sites are closely adjacent, leading to trans modification activity.
More recently, other reports have described the dynamic nature of the interactions between PARP1/ARTD1 and DNA [44], either using its DNA-binding domain (DBD) along with Zinc finger domains I and II (ZI and ZII, respectively) for short term interactions [45], or using histone H4, which leads to a prolonged interaction with the DNA strand [46]. Short term interactions between the DBD domain of PARP1/ARTD1 and DNA were associated with activation of DNA repair pathways at specific stages of DNA damage, while long term interactions between the C-terminal domain of PARP1/ARTD1 and histone H4 were associated with promotion of gene expression [44]. This dual action of PARP1/ARTD1 on DNA illustrates the complexity of this enzyme and provides indications that PARP1/ARTD1 may be associated with chronic inflammation not only as a repair mechanism secondary to inflammation-driven DNA damage [47], but also promoting the expression of pro-inflammatory and/or anti-inflammatory genes.
Reactive oxygen species (ROS) generated during pro-inflammatory responses lead to DNA damage [48], triggering short term PARP1/ARTD1-DNA interactions [44,45]. However, in a study using a model of elastase-induced emphysema and chronic lung inflammation in mice, prolonged inhibition of PARP1/ARTD1 with Olaparib reduced the number of macrophages in the bronco-alveolar lavage after 21 days of treatment when compared with the control group [49]. Levels of ROS and malondialdehyde (MDA, a marker of lipid peroxidation) increased in lung tissues of the control group (four-fold and seven-fold, respectively) due to the inflammation and macrophage activity induced by elastase, but daily treatment with Olaparib restored ROS and MDA to normal levels, indicating an improvement in the inflammatory and redox balances [49,50]. These results exemplify how PARP1/ARTD1 may have a dual and contrasting role in chronic inflammation, repairing DNA following ROS while promoting the production of ROS in macrophages.”
Comment: Lines 282-317 are excessively focused on the mass spectrometry efforts of the authors, completely ignoring a huge body of literature on mass spectrometry analyses of ADP-ribosylation sites by many other groups.
Response: We re-wrote the section discussing mass spectrometry and ADP-ribosylation. As suggested, we included many new references from other research groups, providing a wider view of the development of mass spectrometry in the ADP-ribosylation field. While our articles remain an important part of this section, they are contextualized among other important articles. Please, se below the new version.
Pages 14-17, lines 434-524:
“5. Mass Spectrometry and ADP-ribosylation
Enzyme-catalyzed covalent modifications of amino acids such as phosphorylation, ubiquitination, glycosylation, and ADP-ribosylation are posttranslational modifications (PTMs). PTMs regulate various processes related to cellular homeostasis [125]. The biological complexity and the potentially ephemeral nature of PTMs make them challenging to decipher; but innovative mass spectrometry technologies have enabled their widespread investigations. Ribosylomics is the study of proteome-wide ADP-ribosylation, using mass spectrometry. The last ten years have witnessed technological advances that have overcome initial obstacles for ribosylome, including difficulties associated with precise identification of amino acid acceptor sites, the unstable nature of its covalent binding to the amino acid chain, and its complex pattern of fragmentation [126-132]. Several mass spectrometry-based workflows are currently available to study ADP-ribosylation, but PARylated peptides are not amenable to typical mass spectrometric acquisition methods. The conversion of PAR to MAR peptides using poly-PARG [80,131,132]or the complete reduction to a phosphoribose using a phosphodiesterase [133], provide the means to detect ADP-ribosylated proteins/peptides using mass spectrometry; however, these methods do not provide the nature of the original modification, MAR vs PAR. In parallel to the mass spectrometry-based method innovations to characterize ADP-ribosylated proteins, are the ongoing efforts to develop computational resources to confidently characterize and report ribosylome data.
5.1 Enrichment strategies and activation methods influence the identification of ADP-ribosylated proteins in macrophages
Multiple research groups have tested the activation methods for identifying ADP-ribosylated peptides, their ADP-ribose acceptor sites, and unique enrichment strategies. Electron transfer dissociation (ETD) proved to be efficient in the identification of unambiguous ADP-ribosylated peptides and their acceptor sites [127], being the combination of ETD with higher-energy collisional dissociation (EThcD) superior to ETD alone for the same purpose [128,132]. Also, the enrichment protocol using Af1521-sepharose beads workflow [134] can be combined with ETD for mass spectrometry analysis [131]. Different activation methods may provide the identification of specific ADP-ribosylated peptide groups, depending on their acceptor sites. Ion ETD seems to be superior to EthcD in the occurrence of non-dissociative electron transfer for ADP-ribosylated precursor peptides, and residues modified on arginine and lysine were more stable during HCD fragmentation, whereas the annotation of residues modified on serine, glutamate, tyrosine, and aspartate were more challenging [126]; this is interesting, as modifications on arginine were more frequent during physiological conditions, while modifications on serine were scarce in similar conditions [131], indicating that adjusting the activation method based on the biological condition may provide more reliable results. It is worth noting that the studies mentioned in this paragraph mainly used cancer cell lines and/or healthy mice tissues, indicating that technical optimization would potentially be needed for the study of inflammatory biosystems related to macrophages and macrophage-like cells.
In 2019, our group demonstrated that different ADP-ribosylation enrichment strategies and activation methods influence the identification of ADP-ribosylated peptides in IFN-γ-stimulated human THP-1 macrophage-like cells [80]. We compared the mass spectrometry results of two enrichment strategies: the Af1521-based workflow [134], in which the macrodomain of the Af1521 compound is used to affinity-purify ADP-ribosylated peptides; and the 10H anti-ADP-ribose antibody-based workflow, in which the antibody is used in immunoprecipitation of ADP-ribosylated proteins. While Af1521-based workflow provided spectra rich in MARylated peptides and amino acid acceptor sites, the antibody-based workflow only provides peptides suggesting candidate ADP-ribosylated proteins since ADP-ribosylated peptides themselves are not detected. The 10H strategy enriched 1,389 candidate ADP-ribosylated proteins, whereas the Af1521 strategy enriched 145 ADP-ribosylated proteins, resulting in 39 proteins commonly identified of which included PARP14/ARTD8 and PARP9/ARTD9 [80].
We also compared distinct activation (peptide sequencing) methods and demonstrated that, while HCD provides a larger number of identified ADP-ribosylated peptides, ETD dissociation provides a more reliable identification of the ADP-ribosylation acceptor site in ADP-ribosylated peptides [80]. With these results, we were able to confirm that stimulation of human THP-1 macrophage-like cells with IFN-γ increased PARP9/ARTD9 and PARP14/ARTD8 ADP-ribosylation levels.
5.2 An innovative spectral annotation strategy facilitates the report of ADP-ribosylated peptides in IFN-γ-stimulated mice
The investigation of ADP-ribosylated proteins using mass spectrometry methods requires optimal annotation strategies to accurately identify such proteins after enrichment protocols. The ADPriboDB (initially published by Vivelo et al. [135] and updated by Ayyappan et al. [136]) was the first report of a publicly available database of ADP-ribosylated proteins, in which users can find information about proteins reported in the mass spectrometry literature, from as early as 1975. Each individual entry was revised manually by two independent reviewers before inclusion in the database. Likewise, manual annotation of peptide spectra is still commonly used in mass spectrometry studies on the ADP-ribosylation field.
In 2022, our group capitalized on our optimized ADP-ribosylation enrichment and activation methods to develop a new strategy for annotating ADP-ribosylated peptide spectra (named “RiboMap”) from liver and spleen samples of IFN-γ-stimulated mice [129,137]. In this strategy, once a candidate ADP-ribosylated peptide fragment spectrum is assigned and scored by the standard spectral search engine, RiboMaP then annotates and scores the spectra for ADP-ribosylation-unique features [129].
With this unique spectral annotation tool, we could increase the confidence of the reported ADP-ribosylated peptide spectra associated with pro-inflammatory responses in liver and spleen. With that combination of mass spectrometry and computational techniques, even ADP-ribosylated peptides with overall low biological abundances, such as PARP9/ARTD9 and PARP14/ARTD8, could be identified. We further applied the RiboMap strategy to publicly available data sets and even to our own previously published human macrophage cell and mouse samples, and we found that, regardless of study and sample type, RiboMap increased the number of ADP-ribosylated peptide spectral annotation in all tests [129].”
Comment #3: “Large parts of the manuscript are devoted to describing the results of individual papers, without integrating the knowledge gained from these studies into a coherent narrative, which is arguably the main purpose of a review manuscript. Examples: Lines 123 – 148 (about ref. 39); lines 149-162 (about ref.44); 163-186 (about ref. 45) and section 3.2.”
Response: We implemented many changes in our manuscript to address this comment. Firstly, we re-wrote the excerpts that were focused on a small number of articles or those in which we explored a given article too extensively (please, see the examples below). In this process, we incorporated XX new references to our main text and expanded specific sections to provide readers with a more balanced review of the current literature. In addition to re-writing parts of the manuscript and integrating them, we created a whole new section named “SARS-CoV-2, ADP-ribosylation and innate immune response”, in which we explored host-pathogen interactions involving SARS-CoV-2 and ADP-ribosylation. We also created new subsections focused on host-pathogen interactions, named “PARP9/ARTD9 mediates viral and bacterial host-pathogen interactions” and “PARP14/ARTD8 mediates chronic inflammation and response to arboviruses”. For your convenience, please see below the excerpts mentioned above.
Pages 6-8, lines 179-239:
“2.2. PARP1/ARTD1 promotes transcription of pro-Inflammatory and apoptosis-related genes
Inhibition of PARP1/ARTD1 ameliorates inflammation in chronic conditions and innate immune responses, and this effect was found in multiple pathologies driven by long term inflammatory processes. Kunze et al. [51] demonstrated that stimulation of bone marrow-derived monocytes (BMDMs) from genetically modified mice expressing catalytically inactive PARP1/ARTD1 induced the expression of a pro-inflammatory signature of almost 2,500 genes, including genes regulating IL-12, IFN-γ, and TNF-α production. In the same study, they reported that mice transplanted with catalytically inactive PARP1/ARTD1 myeloid progenitors were colonized by H. pylori at higher levels when compared to their control littermates [51], suggesting that PARP1/ARTD1 contributed to controlling gastric bacterial colonization. In another disease model, inhibition of PARP1/ARTD1 with 3-aminobenzamide, an anti-inflammatory compound classically used for PARP1/ARTD1 inhibition [52], improved rectal hemorrhage, blood sugar levels, blood IL-1β levels, weight loss, and the histological score of colonic sections in mice with colitis associated diabetes [53]. Similar findings were reported by Kovács et al. [54] after using Olaparib to inhibit PARP1/ARTD1 activity in a mice model of Chron’s disease (a type of inflammatory bowel disease). They found that Olaparib increased the levels of IL-10, while suppressed the concentration of IL-1β and IL-6 [54]. Also, Olaparib generated a reduction in the number of monocytes in the blood of treated mice when compared with controls [54]. Gupte et al. [55] stimulated BMDMs from wild-type and PARP1/ARTD1-deficient mice, demonstrating that PARP1/ARTD1-mediated STAT1-α PARylation influenced the transcriptional program upon IFN-γ stimulation [55].
The regulation of PARPs/ARTs catalytic activities in chronic inflammation also relates to NAD+ metabolism. Gerner et al. [56] inhibited nicotinamide phosphoribosyltransferase (NAMPT), a rate-limiting enzyme in the NAD+ salvage pathway, to reduce NAD+ levels in human cells and mice with intestinal colitis. They found that depletion of NAD+ reduced PARP1/ARTD1 catalytic activity, suppressed the expression of IL-6, IL-1β, and TNF-α, and skewed monocytes/macrophages from pro-inflammatory towards anti-inflammatory phenotypes [56]. In the same line, reduction of NAMPT-derived NAD+ via pharmacological inhibition of NAMPT reduced the pathological changes in psoriasis [57] and atopic dermatitis [58] and diminished the expression of pro-inflammatory biomarkers.
In addition to promoting cytokine/chemokine gene expression, PARP1/ARTD1 also influences the cellular fate in apoptosis [59], a fundamental element of inflammation [60]. PARP1/ARTD1 has been extensively associated with caspases in a mechanism known as Parthanatos [61,62] (not reviewed in this manuscript). For instance, Zhang et al. [63] analyzed cleaved caspase-3 in liver samples from mice with chronic alcoholic liver injury [63], and demonstrated that pharmacological inhibition with PJ-34 [64] or genetic depletion of PARP1/ARTD1 decreased the number of cleaved caspase-3-positive cells in diseased livers when compared to controls. They found that long term ethanol consumption promoted PARP/ART activation, hepatic steatosis, and intense cytokine expression in liver samples, while in vivo pharmacologic inhibition of PARP1/ARTD1 with PJ-34 attenuated triglycerides content and serum alanine transaminase levels in liver, suggesting a milder injury phenotype [63]. Erener et al. [65] also identified an association between caspase 1, caspase 7, and PARP1/ARTD1. They found that stimulation with LPS promoted the translocation of caspase 7 to the nucleus (mediated by caspase 1 and NLRP3 inflammasome activation), where it cleaves PARP1/ARTD1 at the caspase cleavage site D214 generating free PARP1/ARTD1 fragments, decondensation of chromatin, and expression of NF-kB dependent-genes. They generated human THP-1 cells expressing non-cleavable PARP1/ARTD1, stimulated them with LPS, and compared them with genetically unmodified controls, confirming that caspase 7 cleaved PARP1/ARTD1 mostly at the D214 site. Martínez-Morcillo et al. [66] found that PARP1/ARTD1 activation leads to skin inflammation and cell death via Parthanatos-mediated apoptosis in psoriasis, and pharmacological inhibition of NAMPT decreased the expression of genes associated with psoriasis.
Together, those findings suggest that PARP1/ARTD1 can influence gene expression during chronic inflammation via ADP-ribosylation of macromolecules and can initiate apoptosis upon interaction with caspases. Controlling NAD+ levels via NAMPT regulation in such immune responses may be a potential source of new targets to suppress pathogenesis derived from ADP-ribosylation, although a deeper understating of these mechanisms is still needed.”
Page 11-12, lines 346-362:
“3.4 PARP9/ARTD9 and PARP14/ARTD8 mediate macrophage activation in atherosclerosis
PARP14/ARTD8 is also important in other chronic inflammatory responses. Using a systems approach based on unbiased network analysis and artificial intelligence, our previous studies discovered PARP14/ARTD8 and PARP9/ARTD9 as potential molecular switches of macrophage activation [79,80]. Proteome analyses from stimulated and non-stimulated human and mouse macrophage-like cells detected an increase in the ADP-ribosylated PARP14/ARTD8 and PARP9/ARTD9 peptide levels upon stimulation with IFN-γ, and network analysis identified a close link between those PARPs/ARTs and the human coronary artery disease gene module [79,80]. Additional in vitro experiments indicated that PARP9/ARTD9 and PARP14/ARTD8 may function upstream of pro-inflammatory STAT1 and anti-inflammatory STAT6 signaling pathways, respectively [79,80]. Iqbal et al. [90] reported that macrophages from PARP14/ARTD8-deficient mice express higher levels of tissue factor mRNA and protein than do wildtype mice [90]. Mehrotra et al. [91] reported that PARP14/ARTD8 specifically binds to STAT6, regulating its promoter activity upon stimulation with IL-4, and demonstrated that this interaction is dependent on PARP14/ARTD8 catalytic domain [91].”
Pages 12-13, lines 364-427
“4. SARS-CoV-2, ADP-ribosylation and innate immune response
ADP-ribosylation and PARPs/ARTs are important in viral host-pathogen interactions and in the organization of host’s innate immune response [92]. PARPs/ARTs genes are interferon-stimulated genes [93,94]. Stress granule formation is a major effect of interferon stimulation during innate immune responses, and it is closely related to PARPs/ARTs and ADP-ribosylation [95-97]. PARPs/ARTs and PARG probably mediate the assembly and maintenance of stress granules in a dynamic way: PARylation of stress granule proteins increases in stress conditions or with PARG silencing [15,98]. Together, these results suggest that hydrolysis of PAR/MAR could limit the effectiveness of the host’s innate immune response against viruses.
Even before the severe acute respiratory syndrome coronavirus (SARS-CoV)-2 pandemic, different types of coronaviruses’ macro domains belonging to the nonstructural protein 3 (nsp3) were identified as ADP-ribose binding modules [99]. In 2006, Egloff et al. reported that the crystal structure of SARS-CoV macro domain associates with ADP-ribose, being able to bind PAR and to function as an ADP-ribose 1”-phosphatase [100]. Evidence suggests that nsp3 and its macro domains were part of coronaviruses’ virulence mechanisms [101-103], promoting virus replication and suppressing interferon-mediated host responses (e.g., stress granules formation) [104-106].
With the onset of the pandemic, data connecting the new SARS-CoV-2 macro domains and ADP-ribosylation quickly became available [107]. The crystal structures of SARS-CoV-2 nsp3 and its macro domains were the initial focus of many research groups, often associating structural studies [108-111] and computational methods [112,113] to identify potential treatments for the infection. Alhammad et al. [114] reported that SARS-CoV-2 nsp3 macro domain 1 (Mac1) hydrolyzes MARylated proteins, functioning as a mono-ADP-ribosylhydrolase. This macro domain function is preserved across the three coronavirus that caused pandemics in a recent past: SARS-CoV, SARS-CoV-2, and Middle East respiratory syndrome coronavirus (MERS). Brosey et al. [115] compared human PARG with Mac1 crystal structures and identified homology between their active sites, revealing that PARG inhibitor fragments PARG-345 and PARG-329 can fully interact with Mac1, posing as potential inhibitors for Mac1. Chea et al. [116] proposed that Mac1 has specific targets and functionality when compared to Mac2 and Mac3. Their results indicate that Mac1 may act specifically in the ADP-ribose moieties on O- and N-linked groups, being able to cleave ADP-ribosylated substrates via a-NAD+, ADPr-1”phosphate, and O-acetyl-ADP-ribose, but not via b-NAD+, a-ADP-ribose-(arginine), and ADP-ribose-(serine)-histone H3.
Other studies also investigated the link between Mac1 and innate immunity responses against SARS-CoV-2. Russo et al. [117] demonstrated that ectopic nsp3 (macro domain not specified) is able to hydrolyze downstream ADP-ribosylation mediated by PARP9-DTX3L dimers following IFN-γ stimulation. Preliminary data indicated [118] that deletion of Mac1 in SARS-CoV-2 (ΔMac1) led to a faster clearance of the virus in a mouse model of severe infection when compared to wild-type SARS-CoV-2. ΔMac1 also promoted the expression of ISGs and interferons, and sharply reduced the number of inflammatory neutrophils and macrophages.
There is another mechanism by which SARS-CoV-2 may intervene in the host-pathogen interaction via ADP-ribosylation. PARP/ART catalytic activity depends on NAD+ for the covalent transference of ADP-ribose to biological macromolecules. It is also well established that increased PARP/ART catalytic activity leads to depletion of NAD+ [119,120]. Reports before SARS-CoV-2 already suggested that restoration of NAD+ would enhance host immune responses against viruses aiding macrophage function and the interferon cascade [121]. Based on this background, authors hypothesized that NAD+ may also be a key element of pathogenesis in acute and chronic (post-acute sequelae of COVID-19) SARS-CoV-2 infection [120,122,123]. Heer et al. [23] demonstrated that varied human lung cell lineages infected with SARS-CoV-2 have increased expression of PARP7/ARTD14, PARP10/ARTD10, PARP12/ARTD12, and PARP14/ARTD8 (among other PARPs/ARTs), and that NAD+ concentration was the limiting factor for these enzymes. In addition, the authors of the same study demonstrated that infection of human cells with murine hepatitis virus (a model of coronaviruses) lead to NAD+ and NADP+ depletion, and that SARS-CoV-2 changes the expression of genes related to NAD biosynthesis [23]. Using SARS-CoV-2 infected mice, Jiang et al. [124] confirmed that SARS-CoV-2 infection alters the expression of genes related to NAD and NADPH biosynthesis. They also demonstrated that NAD+ supplementation alleviated the pathological phenotypes of pneumonia in infected mice, and partially rescued the unbalance in NAD+ genes.”
Page 10-11, lines 310-344
“3.2. PARP9/ARTD9 mediates viral and bacterial host-pathogen interactions
In a cohort with patients infected with pulmonary tuberculosis (TB) and healthy controls, Chen et al. [83] identified an inversely proportional association between TB infection severity and methylation status of PARP9/ARTD9 DNA in PBMCs extracted from participants. Severe TB clinical phenotypes associated with hypomethylation of PARP9/ARTD9 gene, suggesting that lower expression of PARP9/ARTD9 may lead to impaired innate response to TB infection in individuals with that epigenotype. Novel data from Thirunavukkarasu et al. [84] further support this hypothesis. They reported that PARP9/ARTD9 mRNA was increased in humans and mice infected with TB, and Parp9-/- mice were more susceptible to TB infection and developed more severe phenotypes when compared to controls.
Similarly, PARP9/ARTD9 appears to be involved in innate immune responses against RNA viruses. Xing et al. [85] demonstrated that PARP9/ARTD9 is able recognize and bind RNA virus in human and mice dendritic cells and macrophages, deploying an IFN-mediated response independent of the mitochondrial antiviral-signaling (a major mechanism for recognizing RNA viruses during infection). Furthermore, Parp9-/- deletion made mice more susceptible to RNA virus infection [85], reinforcing that PARP9/ARTD9 participates in the host-pathogen interactions. Curiously, PARP9/ARTD9 was associated with persistent Hepatitis B virus (HBV) infection in a transcriptome-wide association study, in which chronic HBV carriers had increased expression of PARP9/ARTD9 when compared to non-infected individuals [86]. HBV is a DNA virus with unique features that approximate it to RNA viruses [87], which may relate to the results above in this paragraph (PARP9/ARTD9 acting as a recognizer of viral RNA).
3.3 PARP14/ARTD8 mediates chronic inflammation and response to arboviruses
Recent data indicate that PARP14/ARTD8 participates in the establishment of an immune response to arboviruses. Eckei et al. [88] reported that the macrodomains of Chikungunya virus (a positive single strand RNA virus) have strong hydrolase activity on proteins that were ADP-ribosylated by PARP10/ARTD10, PARP14/ARTD8 and PARP15/ARTD7. Fernandez et al. [89] reported that Zika virus infection in human PBMCs induced the expression of PARP14, IL-6, CCL8, CXCL1, and CXCL5, suggesting that the infection promoted changes in the transcriptional and post-transcriptional levels. These results indicate that PARP14/ARTD8 influences the host-pathogen dynamic in arboviruses infections.”
Reviewer 4 Report
The manuscript by Santinelli-Pestana. et al., entitled " PARPs and Ribosylation in Chronic Inflammation: A Focus on Macrophages” is an interesting review aimed at ribosylation process in inflammation.
The topic is interesting and the work is clearly justified. Several modifications should be introduced in the text to improve the quality of the manuscript and to make it more clear. I support the publication of the manuscript in Pathogenes after minor revision.
Line 51: Reference is needed on PARP1 in adipogenesis.
Line 58: “PARP biology has been studied in the context of innate immune system, with a particular focus on macrophages (12,13).” The authors cite two of their own works 12 and 13, but it is not clear what is there in these review and research article. It is better to review them in detail.
In Part 2 “PARylation, Macrophages, and Chronic Inflammation” Figure 1 is cited. There are other PARPs on Figure 1 except PARP1, but there is nothing in the text of Part 2 about PARP7 and PARP9/PARP14. The reader can find the information about it in Part 3, but there is no reference on Figure 1 there.
Lines 98-108: It would be nice to give a Figure on structural organization and dual action of PARP1.
Line 110 and Line 161: (as described above) – where was it above?
Line 114: Reference is needed on the influence of PARP1 inhibition with olaparib on the number of macrophages.
Line 118: As previously discussed – it is not clear when it was discussed.
Line 126: Reference is needed.
Line 186: What disease is meant?
Line 197: The authors cite the works 47 and 48, it is better to review them more in detail.
Line 217: It is better to describe Figure 2 in more detail.
Line 232: What does the systems approach mean?
Author Response
Response to Reviewer #4
Comment #1: “The manuscript by Santinelli-Pestana. et al., entitled " PARPs and Ribosylation in Chronic Inflammation: A Focus on Macrophages” is an interesting review aimed at ribosylation process in inflammation.
The topic is interesting and the work is clearly justified. Several modifications should be introduced in the text to improve the quality of the manuscript and to make it more clear. I support the publication of the manuscript in Pathogenes after minor revision.”
Response: Thank you for valuable comments and suggestions. We reviewed our manuscript and promoted several changes based on your comments. Below you will find each comment/suggestion and the respective answer with the modification made to address your concerns. Please, see below.
Comment: Line 51: Reference is needed on PARP1 in adipogenesis.
Response: We included the reference “Poly(ADP-Ribose)Polymerase-1 (PARP1) Controls Adipogenic Gene Expression and Adipocyte Function” by Erener et al (2012) [https://academic.oup.com/mend/article/26/1/79/2614819?login=false] to the manuscript. (Page 3, line 76)
Comment: Line 58: “PARP biology has been studied in the context of innate immune system, with a particular focus on macrophages (12,13).” The authors cite two of their own works 12 and 13, but it is not clear what is there in these review and research article. It is better to review them in detail.
Response: We re-wrote this excerpt and made it clearer what we meant and how the cited works support the statement. The revised paragraph is: “PARP/ART biology has been studied in the context of the innate immune system, with a particular focus on macrophages. PARPs/ARTs were associated with biological responses mediated by IFN-γ, TNF-α, IL-1β, IL-6, and NF-κB, such as host-pathogen interactions in viral infections, vascular inflammation, and others”. (Page 3, lines 82-85)
Comment: In Part 2 “PARylation, Macrophages, and Chronic Inflammation” Figure 1 is cited. There are other PARPs on Figure 1 except PARP1, but there is nothing in the text of Part 2 about PARP7 and PARP9/PARP14. The reader can find the information about it in Part 3, but there is no reference on Figure 1 there.
Comment: Lines 98-108: It would be nice to give a Figure on structural organization and dual action of Response: Considering your constructive comments on our figures, we re-structured the content and the order of our figures. The new Figure 1 now contains a structural and organizational overview of PARP1 actions and was placed in the section discussing PARP1 and PARylation. The content of previous Figure 2 was partially incorporated into the new Figure 1, and partially excluded for the sake of transmitting a concise and clear message. The previous Figure 1 was preserved and renamed as Figure 2; it was moved to the section in which we discuss MARylation, so readers will have a full understanding of the contends being exhibited. Please, see below the new version of Figures 1 and 2, as well as their legends for your reference.
Page 5, line 141:
Figure 1. Graphical representation of PARP1/ARTD1 and its role during inflammation. A: Tridimensional representation of PARP1/ARTD1 structure, indicating the Catalytic, the WGR, and the BRCT domains. B: Flowchart depicting the cell stimuli able to promote PARP1/ARTD1 activity (catalytic and non-catalytic) during prolonged inflammatory processes, leading to PARylation and transcription of mRNA.
Page 9, line 296
Figure 2. Representation of the disease models influenced by the interplay between PARPs/ARTs and macrophages in chronic inflammation.
Response: We removed the term “as described above” from line 110 to avoid confounding readers; in the new version of the manuscript, there is a clear transition between the paragraphs, making the statement “as described above” redundant. Please, see the excerpt below.
Pages 5-6, lines 160-166::
“This dual action of PARP1/ARTD1 on DNA illustrates the complexity of this enzyme and provides indications that PARP1/ARTD1 may be associated with chronic inflammation not only as a repair mechanism secondary to inflammation-driven DNA damage [47], but also promoting the expression of pro-inflammatory and/or anti-inflammatory genes.
Reactive oxygen species (ROS) generated during pro-inflammatory responses lead to DNA damage [48], triggering short term PARP1/ARTD1-DNA interactions [44,45].”
Likewise, the paragraph containing line 161 and the term “as described above” was modified, and the statement was removed.
Comment: Line 114: Reference is needed on the influence of PARP1 inhibition with olaparib on the number of macrophages.
Response: We incorporated the reference “PARP-1 inhibition ameliorates elastase induced lung inflammation and emphysema in mice” by Dharwal (2018) to that statement (page 6, lines 167-170).
Comment: Line 118: As previously discussed – it is not clear when it was discussed.
Response: We re-wrote this paragraph and removed the term “as previously discussed” to avoid confusing our readers. Please, see the excerpt below.
Pages 6-7, lines 170-177::
“Levels of ROS and malondialdehyde (MDA, a marker of lipid peroxidation) increased in lung tissues of the control group (four-fold and seven-fold, respectively) due to the inflammation and macrophage activity induced by elastase, but daily treatment with Olaparib restored ROS and MDA to normal levels, indicating an improvement in the inflammatory and redox balances [49,50]. These results exemplify how PARP1/ARTD1 may have a dual and contrasting role in chronic inflammation, repairing DNA following ROS while promoting the production of ROS in macrophages.”
Comment: Line 126: Reference is needed.
Response: This subsection was re-written, and adequate references were added. Please, see below for your convenience.
Pages 6-7, lines 179-200::
“Inhibition of PARP1/ARTD1 ameliorates inflammation in chronic conditions and innate immune responses, and this effect was found in multiple pathologies driven by long term inflammatory processes. Kunze et al. [51] demonstrated that stimulation of bone marrow-derived monocytes (BMDMs) from genetically modified mice expressing catalytically inactive PARP1/ARTD1 induced the expression of a pro-inflammatory signature of almost 2,500 genes, including genes regulating IL-12, IFN-γ, and TNF-α production. In the same study, they reported that mice transplanted with catalytically inactive PARP1/ARTD1 myeloid progenitors were colonized by H. pylori at higher levels when compared to their control littermates [51], suggesting that PARP1/ARTD1 contributed to controlling gastric bacterial colonization. In another disease model, inhibition of PARP1/ARTD1 with 3-aminobenzamide, an anti-inflammatory compound classically used for PARP1/ARTD1 inhibition [52], improved rectal hemorrhage, blood sugar levels, blood IL-1β levels, weight loss, and the histological score of colonic sections in mice with colitis associated diabetes [53]. Similar findings were reported by Kovács et al. [54] after using Olaparib to inhibit PARP1/ARTD1 activity in a mice model of Chron’s disease (a type of inflammatory bowel disease). They found that Olaparib increased the levels of IL-10, while suppressed the concentration of IL-1β and IL-6 [54]. Also, Olaparib generated a reduction in the number of monocytes in the blood of treated mice when compared with controls [54]. Gupte et al. [55] stimulated BMDMs from wild-type and PARP1/ARTD1-deficient mice, demonstrating that PARP1/ARTD1-mediated STAT1-α PARylation influenced the transcriptional program upon IFN-γ stimulation [55].”
Comment: Line 186: What disease is meant?
Response: The contents of line 186 are: “…ethanol-induced liver injury. Zhang et al found no changes in PARP2 protein levels in…”. Nevertheless, in the surrounding paragraphs we have other mentions to the word “disease”. In “Chagas disease”, disease refers to the pathology caused by the parasite Trypanosoma cruzi. In “disease model”, disease refers to a given pathology or condition that is partially or fully reproduced by an animal model.
Comment: Line 197: The authors cite the works 47 and 48, it is better to review them more in detail.
ResponseWe re-wrote the subsection discussing Chagas Heart Disease, in which we deepened the discussion on references 47 and 48, as well as incorporated new references. Please, see below for your convenience.
Page 8, lines 241-264:
“2.3. PARP1/ARTD1 mediates host-pathogen interactions in Chagas Heart Disease
Chagas Heart Disease is caused by the protozoan parasite Trypanosoma cruzi (T. cruzi). The classical phenotype seen in this condition is the result of chronic (years to decades) of sustained myocyte inflammation, oxidative stress, and macrophage infiltration into cardiac muscle [67,68]. Ba et al. [69] demonstrated that T. cruzi infection of cardiomyocytes leads to mitochondrial production of ROS that diffuse to the cytosol and nucleus leading to DNA damage and PARP1/ARTD1 activation. As a result, the expression of genes related to pro-inflammatory cytokines increased either due to the interaction between ROS and cytosolic NF-kB or due to PARP1/ARTD1-mediated PARylation of proteins that interact with RelA(p65) (an NF-kB subunit). Further evidence indicated that depletion of PARP1/ARTD1 (with genetic deletion or PJ-34 administration) in infected mice prevented cardiac hypertrophy and left ventricle dysfunction and restored the mitochondrial antioxidant/oxidant balance [70]. PARP1/ARTD1 associated with chromatin during T. cruzi infection but its mRNA levels did not change when compared to non-infected states, indicating that a translocation of PARP1/ARTD1 to chromatin dense regions occurred [71,72]. These results suggest that PARP1/ARTD1 influences the response to mitochondrial stress during T. cruzi infection. Evidence also connects PARP1/ARTD1 to macrophages in the host-pathogen interaction. Macrophage-like RAW264.7 cells treated with extracellular vesicles (EVs) derived from infected mice plasma released higher levels of TNF-α, IL-1β, and IL-6 than did control cells [73]. EVs derived from T. cruzi-infected RAW264.7 cells induced lower expression levels of TNF-α, IL-1β, and IL-6 in BMDMs harvested from PARP1/ARTD1-deficient mice compared to wildtype control [73]. Thus, it is possible that the previously described role of macrophages in Chagas Heart Disease [74] may be mediated by PARP1/ARTD1, but more studies are needed.”
Comment: Line 217: It is better to describe Figure 2 in more detail.
Response: Please, see the answer above on restructuring of the figures and their contents.
Comment: Line 232: What does the systems approach mean?
Response: We re-wrote the statement, to make the meaning of “approach” more understandable. Please, see below for your convenience.
Page 11, lines 347-350:
“PARP14/ARTD8 is also important in other chronic inflammatory responses. Using a systems approach based on unbiased network analysis and artificial intelligence, our previous studies discovered PARP14/ARTD8 and PARP9/ARTD9 as potential molecular switches of macrophage activation [79,80].”
Round 2
Reviewer 2 Report
The authors appreciated most of comments and suggestions and, following them, the paper was improved a lot. Thus, it is my opinion that it is now acceptable for publication.
Reviewer 3 Report
The manuscript was comprehensively and adequately revised and the authors have made large strides in addressing the points I raised. It is acceptable for publication in Pathogens now.